

# The effect of clouds and precipitation on the aerosol concentrations and composition in a boreal forest environment

Sini Isokääntä[1], Paul Kim[2], Santtu Mikkonen[1,3], Thomas Kühn[1,4,*], Harri Kokkola[4], Taina Yli-Juuti[1], Liine Heikkinen[5,6], Krista Luoma[5], Tuukka Petäjä[5], Zak Kipling[7], Daniel Partridge[2] and Annele Virtanen[1]

[1]Department of Applied Physics, University of Eastern Finland, Kuopio, 70210, Finland

[2]College for Engineering, Mathematics, and Physical Science, University of Exeter, Exeter, EX4 4QF, United Kingdom

[3]Department of Environmental and Biological Sciences, University of Eastern Finland, Kuopio, 70210, Finland

[4]Atmospheric Research Centre of Eastern Finland, Finnish Meteorological Institute, Kuopio, 70211, Finland

[5]Institute for Atmospheric and Earth System Research (INAR) / Physics, Faculty of Science, University of Helsinki, Helsinki, 00014, Finland

[6]Department of Environmental Science (ACES) and Bolin Centre for Climate Research, Stockholm University, Stockholm, 10691, Sweden.

[7]European Centre for Medium-Range Weather Forecasts, Reading, RG2 9AX, United Kingdom

[*]now at: Weather and Climate Change Research, Finnish Meteorological Institute, Helsinki, 00101, Finland

*Correspondence to*: Sini Isokääntä (sini.isokaanta@uef.fi)

## Abstract

Atmospheric aerosol particle concentrations are strongly affected by various wet processes, including wet scavenging (below and in-cloud) and aqueous phase oxidation in-cloud. This study employs airmass history analysis and observational data to investigate how wet scavenging and cloud processes affect particle concentrations and composition during transport to a rural boreal forest site in northern Europe. Long term particle size distribution (~ 15 years) and composition measurements (~ 8 years) were utilized in combination with airmass trajectories with relevant variables (e.g. rainfall rate, relative humidity, mixing layer height) from reanalysis data. Additional observational data sets (e.g. temperature, trace gases) were used to further evaluate the wet processes along trajectories with mixed effects models.

All investigated chemical species (sulfate, black carbon and organics) showed an exponential decrease in the particle mass concentration as a function accumulated precipitation along the airmass route. Clear seasonal differences in wet removal were observed for sulfate (SO₄) aerosols, whereas organics (Org) and black carbon (eBC) showed more minor differences. The removal efficiency varied slightly among the different reanalysis datasets (ERA-Interim and GDAS) used for the





trajectory calculations, due to the difference in the average occurrence of precipitation events along the airmass trajectories
between the reanalysis datasets.

Aqueous phase processes were investigated by using a proxy for airmasses travelling inside clouds. Significant increases in
total $SO_4$ mass concentration were observed for airmasses recently been inside non-precipitating clouds when compared to
airmasses that had no experience of clouds or precipitation during the last 24 hours before arrival to SMEAR II.

Mixed effects model, in which other contributing factors (e.g. trace gases, local meteorology, diurnal variation) affecting
particle mass concentrations in SMEAR II were considered, also indicated in-cloud production of $SO_4$. Aqueous phase
formation of $SO_4$ was observed despite the reanalysis dataset used in the trajectory calculations. Investigation of the particle
size distribution measurements revealed that most of the $SO_4$ formed in-cloud can be attributed to particle sizes larger than
200 nm (electrical mobility diameter). No significant formation of aqueous phase secondary organic aerosol (aqSOA) was

observed.



# 1 Introduction

Atmospheric aerosol particle concentrations are governed by their sources and sinks. Scavenging of aerosol particles by cloud droplets, ice crystals and precipitation i.e. wet scavenging, is one of the most essential aerosol particle removal processes in the atmosphere. Therefore, a detailed understanding of wet scavenging is necessary e.g. for atmospheric models

to better simulate particle size distribution, aerosol burden and long-range transport, especially to remote (e.g. arctic) areas. Aerosol wet scavenging can be distinguished into in-cloud scavenging, where the particles activate to cloud droplets or ice crystals (nucleation scavenging) which can further collide with interstitial aerosol (impaction scavenging) and are then removed by precipitation, and below-cloud scavenging, where aerosol particles are collected through collisions with falling raindrops and removed from the air (e.g. Ohata et al., 2016). Below-cloud scavenging is an efficient removal process for

ultrafine and coarse particles, whereas in-cloud scavenging is the most important sink for accumulation mode particles (e.g. Andronache, 2003; Textor et al., 2006; Croft et al., 2009; Ohata et al., 2016).

The below-cloud scavenging rate is affected by the rainfall intensity as well as the collection efficiency which is controlled by both the particle and droplet size (e.g. Leong et al., 1983; Andronache, 2003; Chate et al., 2003) as well as the type of precipitation (Andronache et al., 2006; Paramonov et al., 2011). The below-cloud collection efficiency is the fraction of

collected particles of diameter $d_p$ contained within a collision volume of a drop having diameter $D_p$. The collision between aerosol particles and rain droplet is defined by Brownian diffusion, interception and impaction processes (e.g. Bae et al., 2012). The efficiency of below-cloud wet scavenging is often described by the scavenging coefficient. It is defined as the fraction of aerosol particles captured by raindrops per unit of time and is typically calculated from ambient observations before and after a precipitation. A number of studies have determined aerosol scavenging coefficients for various particle

sizes under various rainfall rates (e.g. Nicholson et al., 1991; Andronache, 2003; Laakso et al., 2003; Blanco-Alegre et al., 2018).

The in-cloud scavenging efficiency is controlled by nucleation (i.e. aerosol activation) and impaction scavenging. It is dominated by activation of aerosol particles into cloud droplets (e.g. Ohata et al., 2016) and hence depends strongly on the updraft velocities at cloud base which along with the properties of the aerosol size distribution and the growing cloud droplet

population governs the supersaturation conditions realised close to cloud base (Dusek et al., 2006; Partridge et al., 2012). If supersaturation conditions are well constrained from in-situ observations, the process of particles activating into cloud droplets can be relatively well described by current droplet activation parameterisations (e.g. Abdul-Razzak and Ghan, 2000; Nenes and Seinfeld, 2003; Fountoukis and Nenes, 2005) especially for a basic inorganic chemical species, e.g. sea salt. However, still large uncertainties exist regarding the role of chemical composition in droplet formation (e.g. Lowe et al.,

2019), and further constraining is needed as particle chemical composition is also one of the key factors in droplet formation (Duplissy et al., 2011; Wu et al., 2013; Pajunoja et al., 2015; Väisänen et al., 2016). After the activation, impaction scavenging between the interstitial particles and cloud droplets is also occurring within clouds but it influences sub-micrometer particle concentrations relatively little (Croft et al., 2010).





Aerosol particles are not only scavenged by clouds but also their mass and properties can change due to aqueous phase
processes. Sulfate production caused by aqueous phase oxidation of gaseous sulfur dioxide which condenses onto particles
(e.g. Barth et al., 2000; Ervens, 2015) is considered to be one of the most important mass addition pathways inside clouds
(e.g. Harris et al., 2014; Ervens, 2015 and references therein). It has been estimated that in-cloud oxidation of sulfur might
contribute significantly (∼60−90 %) to the global sulfate budget (Ervens, 2015). The production of secondary organic
aerosol through aqueous phase processes (aqSOA) has been also reported (e.g. Ervens et al., 2011; El-Sayed et al., 2015;
Ervens et al., 2018; Mandariya et al., 2019), and it has been suggested that aqSOA formation is comparable in magnitude
with SOA formation through gas phase oxidation processes (Ervens et al., 2011). The observations of in-cloud (or fog)
formation of new aerosol mass exist (e.g. Sorooshian et al., 2006; Sorooshian et al., 2007; Wonaschuetz et al., 2012; Xie et
al., 2015; Gilardoni et al., 2016; Xue et al., 2016) but they are scarce especially in areas with relatively low pollution levels.
Few experimental studies have combined the information of chemical composition or hygroscopicity (i.e. ability of a particle
to take up water) with both in-cloud and below-cloud wet scavenging to investigate if differences in composition cause
variation in the wet scavenging efficiency of the particles. Chate et al. (2003) obtained the washout coefficients for heavy
rain for 0.02-10 µm particles having different chemical composition with a theoretical approach following the presentation
given by Slinn (1983). In addition, Chate and Devara (2005), observed order of magnitude differences for the collision
efficiencies between the particles and raindrops of various sizes for selected chemical compositions during thunderstorm and
non-thunderstorm precipitation events. Wang et al. (2021) continued with the topic in a modelling study by investigating the
effect of rainfall intensity and type for different aerosol species. They observed no differences in the wet scavenging
efficiency between different rainfall intensities for different aerosol species, but noted that higher rainfall intensities were
needed for larger particles to acquire the same removal efficiency over the tropics. Xu et al. (2020) included airmass origins
into their study of hygroscopicity and chemical composition of aerosols in Mace Head, on the coast of Ireland, and found out
that wintertime aerosols were usually externally mixed for both continental and marine airmasses.
The estimation of the scavenging coefficients is often Eulerian (see e.g. Zhang and Chen, 2007 for definitions of Eulerian
and Lagrangian approaches), as e.g. in Wang et al. (2021), Chate and Devara (2005) and Chate et al. (2003), and based on
the local precipitation measurements or modelled quantities. A Eulerian approach does not consider that the airmasses
arriving to the measurement site have most likely experienced rain during their transport history, thus altering the particle
population during en-route. In addition, particle composition and number- and mass concentration may be highly dependent
on the airmass source area. Alternatively, a Lagrangian approach has a key advantage over Eulerian methodologies in that
individual particle trajectories are employed to allow for a consideration of the effects of airmass history on the aerosol.
Relatively few Lagrangian aerosol-precipitation history studies have been performed. Tunved et al. (2013) reported that
airmasses arriving from central Europe and Russia into the arctic measurement site (Zeppelin station, Ny Ålesund, Norway)
had a relatively high particle mass concentration in all seasons when compared to airmasses from other regions. They also
investigated how precipitation during transport to Zeppelin influenced the local particle population and exhibited an





exponential decrease in submicron particle mass as a function of accumulated precipitation along the airmass trajectories. They suggested that in-cloud scavenging, which is more efficient for larger particles, was the dominant removal process, and thus the largest particles, which have the largest mass, are first removed, followed by smaller particles. Kesti et al. (2020)

investigated the effect of precipitation on the particle size distribution along airmass trajectories as they travel over the Indian ocean to the Maldives. They observed that a greater reduction in the accumulation mode particle concentration usually coincided with precipitation along the trajectory. A recent study investigated how precipitation along airmasses affects aerosol mass and volume observed in Bermuda (Dadashazar et al., 2021). They concluded that remote marine boundary layer aerosol characteristics are relatively sensitive to the precipitation along the airmass trajectories. All these

studies observed clear changes in the aerosol population (either mass or number concentration) due to the precipitation along the airmass route. Both Tunved et al. (2013) and Kesti et al. (2020) concluded that the particles in the accumulation mode size range show strongest sensitivity to the precipitation along the airmass trajectories. Dadashazar et al. (2021) observed strongest sensitivity of the $PM_{2.5}$ mass to accumulated precipitation of up to 5 mm, while accumulated precipitation exceeding this limit had only minor effects to the $PM_{2.5}$ mass. Similar behaviour was described by Tunved et al. (2013) – the

particle number size distribution was clearly affected by up to 10 mm of accumulated precipitation and a horizontal asymptote was achieved beyond that.

To explore the influence of below-cloud scavenging during transport on observed aerosol size distribution and chemical composition in biogenically dominated environment, we utilize here nearly a 15-year long aerosol dataset from the boreal forest station, SMEAR II, including continuous particle size distribution observations, and almost 8 years of particle

composition measurements. These in-situ observations are combined with airmass trajectories calculated from the HYSPLIT trajectory model (Stein et al., 2015) driven by various reanalysis datasets to investigate how the local aerosol population is affected by various wet processes the aerosols experience during their route to SMEAR II. Our main objectives can be summarized into the following three research questions:

1. How efficiently are different chemical species removed from the atmosphere by precipitation?

2. How does the aqueous phase processing taking place in clouds alter the particle mass concentration and composition?

3. If in-cloud formation of new particle mass is observed, what is the size range this mass is distributed in?



## 2 Data and methods

### 2.1 Observations at SMEAR II, Hyytiälä, Finland

Our observational data includes long-term measurements of various aerosols, gases and meteorological variables collected in the SMEAR II (Station for Measuring Ecosystem–Atmosphere Relations: Hari and Kulmala, 2005) station in Hyytiälä, southern Finland. The majority of the data measured in SMEAR II is publicly available at an online database (Junninen et al., 2009, https://smear.avaa.csc.fi/). The station is classified as a rural measurement station surrounded by mostly homogeneous Scots pine (*Pinus Sylvestris*) forest as there are no significant pollution sources nearby. The closest larger city

is Tampere which has  238,140 inhabitants  (Statistics Finland, 2019), located ca. 50 km southwest from SMEAR II. The particle number size distributions were measured with a Differential Mobility Particle Sizer (e.g., Aalto et al., 2001), and our study covers the years from January 2005 to August 2019. The observations cover the size distribution between 3 and 1000 nm (electrical mobility equivalent particle diameter). Mass concentrations for the various size classes were calculated by assuming the particles were spherical and had a constant density of $\rho = 1$ g cm$^{-3}$.

The chemical composition of the particulate matter at SMEAR II were acquired with aethalometer (e.g. Drinovec et al., 2015) and Aerosol Chemical Speciation Monitor (ACSM:  Ng et al., 2011).  The equivalent black carbon (eBC, Petzold et al., 2013) mass concentration data were calculated for the time between July 2006 to August 2019 from aethalometer (AE31 for 2006 – 2017 and AE33 for 2018 – 2019) measurements, which provide absorption coefficients for various wavelengths. The eBC utilized here was derived from the absorption coefficient measured at $\lambda = 880$ nm (as e.g. in Singh et al., 2014;

Helin et al., 2018).
AE31 data, that is not automatically corrected for filter loading effects like AE33 data, was corrected with the algorithm suggested by Virkkula et al. (2007). The cut-off diameter for the eBC measurements was 10 µm. However, as most of the absorbing particulate matter at SMEAR II falls in the submicron range, the eBC measured for $PM_{10}$ is only 10 % higher compared to $PM_1$ measurements (Luoma et al., 2019). Measurements from the ACSM instrument provided the bulk

chemical composition of sub-micron particulate matter, being most efficient at measuring between ~75-650 nm (vacuum aerodynamic diameter), allowing particles up to 1 µm through with less efficient transmission (Liu et al., 2007).  Previous studies, e.g. Chen et al. (2018) have highlighted that hygroscopic growth leads to a shift in the size of dry particles cut off by impactors during sampling. However, this issue is not relevant for these measurements as the cut size of the virtual impactor used at the inlet for ambient air was clearly larger (2.5 µm) than the upper limit of the ACSM measurement range, and after

the virtual impactor, the aerosol was dried before entering the ACSM  (Heikkinen et al., 2020). The data from the ACSM in this study extends from March 2012 to August 2019, including the mass concentrations (µg m$^{-3}$) of total organic (Org), ammonium, ($NH_4$), sulfate ($SO_4$), nitrate ($NO_3$), and chloride (Chl). More details from the ACSM measurements and data treatment can be found from Heikkinen et al. (2020).





Other investigated gas phase variables included concentrations of gaseous nitrogen oxide ($NO_x$, in a unit of ppb), sulfur
dioxide ($SO_2$, ppb), ozone ($O_3$, ppb) and carbon monoxide (CO, ppb). Variables describing the local meteorological
conditions measured were air temperature ($T$, °C), atmospheric pressure at ground level ($p$, hPa), relative humidity (RH, %),
precipitation (liquid water equivalent, $rain_{local}$, mm h$^{-1}$) solar radiation (SolR, W m$^{-2}$), wind speed (WS, m s$^{-1}$) and wind
direction (WD, °). Data coverage, summary statistics and list of the measurement instruments are shown in Tables S1-S3. All
investigated variables are measured near ground level, below the tree canopy.

The original time resolution for each observational variable varies depending on the measurement instrument. Thus, each
investigated variable was averaged into one hour means. All available observational data overlapping with the trajectories
released every hour was investigated (January 2005-August 2019). Data points coinciding with reported wind direction
between 120° and 140° were removed from the data set to exclude the influence from two nearby sawmills reported as major
sources of VOCs and Org (e.g. Liao et al., 2011; Heikkinen et al., 2020). In addition, data rows for which the airmass back-
trajectory crossed the Kola peninsula (for the sake of data-analysis, we used a rectangular box with coordinates of 31-42° of
longitude and 66-70° of latitude to estimate the geographical area of Kola Peninsula), were excluded from the analysis due to
high pollution caused by industry in that area (e.g. Kulmala et al., 2000; Riuttanen et al., 2013; Heikkinen et al., 2020) as this
strong $SO_4$ source could cause significant biases to our analysis. Further data-analysis was conducted in R Statistical
Software and Python, and colour maps for the figures considering colour vision deficiencies were inspired by Crameri et al.
(2020).

## 2.2 Trajectory calculations and airmass source analysis

4-day (96 h) back trajectories were obtained using version 5.1.0 of the HYSPLIT (Hybrid Single-Particle Lagrangian
Integrated Trajectory, Stein et al., 2015) model for the period from January 2005 to August 2019. 4-day long trajectories
were selected, as that is typically long enough period so that even the slow moving airmasses have enough time to travel
from Atlantic and marine areas over to the boreal environment. The arrival height of the trajectories was set to 100 metres
above ground level at the measurement station in SMEAR II. ERA-Interim reanalysis meteorology at 1 degree resolution
was used as the input for calculating the trajectories which were released every hour leading to 24 trajectories per day
(128,520 in total). In addition, reanalysis dataset of GDAS (1 degree resolution, https://www.ready.noaa.gov/archives.php)
was used to further validate our conclusions obtained with the trajectories based on ERA-Interim reanalysis data.

The observational data has been temporally collocated with the airmass trajectory release times. Any measured variable
extending past August 2019 has not been used in this study even if available as ERA-Interim reanalysis meteorological input
has been superseded with ERA5 after that. Variables provided by HYSPLIT along each trajectory are also used in this study
(in addition to the airmass route coordinates), namely the height of the airmass, rainfall rate at the surface (used as a proxy
for the experienced precipitation by the airmass), relative humidity in the airmass and mixing layer height (MLH) for the
current horizontal location of the air mass. The MLH provided from HYSPLIT at SMEAR II was used to estimate the actual



MLH due to absence of local long-term measurements of MLH at the site. Precipitation events along the trajectory are relatively evenly distributed along the 96 hours (Figure S1), having slightly lower occurrence 12-18 hours before the airmasses reach SMEAR II. Locally measured (surface) precipitation values agree relatively well with the estimate from HYSPLIT (Figure S2).

For the statistical model analysis used to support our findings, the airmass trajectories were clustered into source areas by kmeans-clustering, in which the trajectories are partitioned into $k$ clusters and for each cluster a centroid is defined (e.g. Kaufman and Rousseeuw, 1990). Each trajectory is then allocated to the nearest cluster, providing us with geographical source areas for the airmasses to be used as random effects in the mixed effects model (Section 2.3). Clustering was performed using the R Statistical software with the help of the cluster-package (Maechler et al., 2019; R Core Team, 2019)

using the Hartigan-Wong algorithm (Hartigan and Wong, 1979). Other clustering techniques were tested (e.g. partitioning around medoids with different distance metrics), but kmeans provided distinct enough clusters for our purposes. The appropriate number of clusters was determined by evaluating the interpretability of the clusters and inspecting the total within sum of squares (WSS) for different number of clusters in which the "knee" of the WSS curve (indicating smallest dissimilarities within clusters) could indicate the number of clusters (3 to 6 in our case). The final clusters, i.e. source areas

are show in Figures S3 and S4. The statistical model showed no strong sensitivity towards the number of clusters, i.e. same conclusions could be drawn with 4, 5 and 6 clusters.

## 2.3 Statistical mixed effects model

### 2.3.1 General description of the multivariate mixed effect model

Multivariate mixed effects models were used to investigate the significance of various processes affecting the particle

concentrations at the SMEAR II site. Mixed effects models were used as they estimate the variance-covariance structure of the data in addition to the mean of the response variable, and are better justified for grouped data sets with possible hierarchical structures (as e.g. in this study, by airmass sources, months, hour of the day etc.) than fixed effect models (Mehtätalo and Lappi, 2020).  In addition, statistical mixed effects models are an effective tool when interactions between variables are investigated (see e.g. Mikkonen et al., 2011). For example, a study from Yli-Juuti et al. (2021) used a linear

mixed effects model to distinguish the direct/real effect of temperature from other variables affecting the concentration of organic aerosols when investigating the organic aerosol driven climate feedback in the same boreal area. Linear mixed effects model can be presented in general form as

$$\boldsymbol{y} = \mathbf{X}\boldsymbol{\beta} + \mathbf{Z}\boldsymbol{b} + \boldsymbol{\epsilon}, \tag{1}$$

where $\boldsymbol{y}$ is the vector of the response variable, $\boldsymbol{\beta}$ and $\boldsymbol{b}$ are the vectors of fixed and random effects, respectively and $\mathbf{X}$ and $\mathbf{Z}$

are the related design/coefficient matrices (McCulloch et al., 2008). Vector $\boldsymbol{\epsilon}$ includes the random errors. Depending on the structure of the random effects (crossed or nested effects), the relationship between $\mathbf{X}$ and $\mathbf{Z}$ varies (McCulloch et al., 2008).





In our study, we also needed to consider the observed exponential dependency between the response variables and the accumulated precipitation (see Section 3.1) and thus we used a nonlinear mixed effects model. The nonlinear mixed effects models (separate model for each chemical species) were applied with R statistical software (R Core Team, 2019) with *nlmer*-
function provided by the package lme4 (Bates et al., 2015). The formulation of the final fitted equation is expressed as

$$[\text{VAR}_i] = \beta_0 + \{\boldsymbol{b_h} + \boldsymbol{b_m} + \boldsymbol{b_y}\} + \{\beta_1[\text{NO}x_i] + \beta_2[\text{SO}_{2,i}] + \beta_3[\text{O}_{3,i}] + \beta_4[\text{CO}_i]\} + \{\beta_5\text{T}_i + \beta_6[\text{MLH}_i]\} +$$
$$\{\exp(\beta_7\text{accum. precip}_i) + \beta_8\text{time. in. cloud}_i\} + \{\beta_9\text{emission. col. time}_i + \beta_{10}\text{time. in. land}_i + \boldsymbol{b_a}\}, \quad (2)$$

where [VAR] is now the mass concentration of either Org, SO$_4$ or eBC, $\beta_0$ is a model intercept, $\boldsymbol{b_h}$, $\boldsymbol{b_m}$, $\boldsymbol{b_y}$ and $\boldsymbol{b_a}$ are the vectors of random intercepts for hour of the day, month, year and airmass source area, respectively, and $\beta_1$ - $\beta_{10}$ are the fixed
regression coefficients. Subscript *i* denotes the time point i.e. one observation. Thus the predictor variables (see Section 2.1 for the abbreviations) include concentrations of SO$_2$, CO, NO$_x$ and O$_3$ (trace gases); air *T* and MLH (at SMEAR II derived from the back-trajectory data) describing the local meteorology and following trajectory-derived variables: accumulated precipitation along the trajectory (mm), time spent in high humidity conditions without simultaneous rain ("in non-precipitating cloud", h), emission collection time (time in mixed layer until rain event, h) and total time the airmass has spent
over land (h). In addition, the airmass source areas (obtained by clustering as explained in Section 2.2, visualised in Figures S3 and S4) and observation year, month and hour of the day were included, as shown in Eq. (2). Summary of the used predictor variables in the regression is shown in Table 1 and each predictor variable group is separated with curly brackets in Eq. (2). The process leading to the selection of the response variables is explained in the Section 2.3.2.

**Table 1 Predictor variables used in regression.**

| Group | Name | Variables included |
|---|---|---|
| 1 | Base variability (diurnal, seasonal, random) | Observation year and month, hour of the day |
| 2 | Trace gases | NO$_x$, SO$_2$, O$_3$, CO |
| 3 | Local meteorology | T, MLH |
| 4 | Wet processing along the trajectory | Accumulated precipitation, time spent in non-precipitating cloud |
| 4a | Wet scavenging | Accumulated precipitation |
| 4b | In-cloud aqueous phase processing | Time spent in non-precipitating cloud |
| 5 | Long-range transport | Airmass source area, emission collection time, time spent above land |



### 2.3.2 Selection of relevant variables and determining the relative contribution of variable groups

In this section we justify our decision to leave out some of the variables, that could be considered relevant for the response variables (Org, eBC, SO4) we investigated. Variables that were investigated but were excluded from the final model were
RH, SolR, WS, WD, $rain_{local}$ and $p$. RH has strong correlation coefficient ($> 0.5$) with MLH and having both in the model violates the assumption of the model on relatively independent predictors and causes collinearity issues on the computation (e.g. Dormann et al., 2013). When comparing models with either RH or MLH, the models with MLH had better predictive capability, thus MLH was selected. MLH also has high correlation coefficient SolR, but again the model with MLH had better predictive capability and thus SolR was discarded from further analysis. In addition, including both, RH and SolR, had
negligible effect on goodness-of-fit indicators or did not change the other regression coefficients significantly indicating their presence in the model does not improve the overall fit. WS, WD, $rain_{local}$ and $p$ were not significant predictors for the models based on a likelihood ratio test (Wilks, 1938) made between different model versions. Simultaneous observations of the selected variables were used in the final regression, including 22,778 observations in total from the period between March 2012 and August 2019. The division into two seasons based on monthly median temperature $T_m$ (discussed in detail in
Section 3.1) led to 14,501 observations for warm months ($T_m > 10$ °C) and 8,277 observations for the cold months ($T_m < 10$ °C).

To investigate how strong and/or significant an effect each predictor group shown in Table 1 has for the observed variable (Org, eBC, SO$_4$) relative to all other predictor groups, we applied Bayesian Information Criterion (BIC, Schwarz, 1978) derived for each of the fitted models. BIC is a criterion which can be used in model selection as models with lower BIC are
preferred (Schwarz, 1978). It is based on likelihood-function and includes a penalty term for the number of variables in a model to avoid overfitting (Schwarz, 1978; Stoica and Selen, 2004). Each variable group was removed in turn from the full model (separate models for each species) presented in Eq. (2), and the BIC for the reduced model was compared to the BIC of the full model. With this approach, we were able to determine the relative contribution of each variable group for the investigated species (response variable). Regression coefficients and relative contributions are reported in Section S3.



## 3 Results and discussion

### 3.1 Effect of wet scavenging on the aerosol concentrations

The evolution of the total aerosol mass (assuming unit density, 1 μg m$^{-3}$) and number concentration derived from the DMPS size distribution as a function of accumulated precipitation along the airmass trajectories are shown in Figure 1. The hourly rainfall values at the surface (mm h$^{-1}$) provided by the HYSPLIT trajectory data were integrated over the 96-hour period for each trajectory to acquire the total accumulated precipitation during each trajectory at SMEAR II. The accumulated precipitation was then grouped into 0.5 mm bins, and for each bin median particle mass was calculated. Bins which had less than 10 data points were discarded due to low statistics (and thus not shown in Figure 1). The sample size for each bin corresponding to Figure 1 is demonstrated in Figure S5. The particle mass derived from both DMPS and ACSM measurements (Figure S6, corresponding sample size in Figure S7) show exponential decrease (as a function of accumulated precipitation) similarly to the results reported by Tunved et al. (2013) for arctic aerosols. The total particle number (Figure 1b) also shows a decrease in the concentration, but not as clear exponential decrease as shown for the particle mass concentration. The behaviours of the particle mass and number as a function of accumulated precipitation do not depend on the choice of reanalysis data used to drive the HYSPLIT trajectory model (Figure S20).

It should be noted here that this type of approach to air mass history analysis in which the vertical trajectory position with respect to cloud is not considered, does not allow us to explicitly separate the in-cloud (particles activate to cloud droplets and collide with interstitial aerosol and then precipitate) and below-cloud (falling droplets collide with particles) precipitation scavenging. Instead, it gives us an estimate of the overall effect of precipitation on aerosol concentrations by using the surface precipitation provided by the airmass trajectories as a proxy for the precipitation experienced by each single particle trajectory. In addition, as we investigate aerosol scavenging in a Lagrangian framework (visualized in Figure 2) in which the temporal and spatial scales of the reanalysis data used in the trajectory calculations are much larger than dynamic cloud processes, we cannot directly probe sub-grid scale processes, e.g. in-cloud aerosol scavenging. Accordingly, the Lagrangian analysis demonstrates in-cloud scavenging occurring via the removal of activated aerosol particles from the atmosphere due to precipitation scavenging, only if the clouds precipitate while the air parcel is passing them. Therefore, when we investigate how the size distribution changes with accumulated precipitation as demonstrated in Figure 3, some qualitative conclusions can be drawn. The strong exponential decay of particle size distribution is visible in sizes larger than 100 nm while the changes in size range around 10-50 nm are small or negligible. This indicates that the large particles ($d_p >$ 100 nm) are removed most efficiently with the first 10 mm of accumulated precipitation while smaller particles remain unaffected by any amount of accumulated precipitation. Hence, the in-cloud scavenging in our investigation is greatly dependent on the activation of aerosol particles to cloud droplets, which in turn is strongly dependent on the particle size. The number concentration of particles with diameter larger than 100 nm has been widely used as a proxy for aerosol able to activate to cloud droplets. This is also the size range where we see the largest decrease in number concentrations (Figure 3)





as a function of accumulated precipitation. We can qualitatively explain the observed behaviour in Figure 3 according to a simplistic view of the complex and highly dynamical process of activation in which the available supersaturation varies temporally and spatially during the cloud precipitation cycle: after the larger particles are removed through activation into
cloud droplets of which a subset will precipitate, the size of the smallest activated particles decreases because of less competition for the available supersaturation. On the other hand, the lowest scavenging efficiency values for below-cloud scavenging are typically in the size range of some hundreds of nanometres depending on the precipitation type (e.g. Wang et al., 2010) and at size range below 100 nm, the scavenging efficiency increases strongly with decreasing particle size so that at 10 nm size range, it is significantly higher. Hence, if the below-cloud scavenging would play a major role at sub-micron
size range considered here, we should see a decrease in the number concentration with accumulated precipitation in the smallest particle sizes where the below-cloud scavenging efficiencies are the highest. As shown in Figure 3, the aerosol concentrations in size range of 10-50 nm show no sensitivity to accumulated precipitation and the largest decrease in concentrations are shown in size range of $d_p > 100$ nm suggesting that the in-cloud scavenging is the dominating removal mechanism in the submicron particle size range in the studied environment. This has further support from earlier studies
suggesting below-cloud scavenging to be a less important scavenging mechanism than in-cloud scavenging for accumulation mode sized particles (e.g. Tunved et al., 2013; Wang et al., 2021). Similar changes in the size distribution can be observed when the analysis was repeated using GDAS reanalysis meteorology instead of ERA-Interim (Figure S21).

Assuming now in-cloud scavenging to be dominating and referring back to Figure 1, the difference in the decreasing trends between particle mass and number concentration arises likely from the fact that the aerosol mass is dominated by large ($d_p >$
100 nm) particles, which are more efficiently removed by in-cloud wet scavenging when compared to particles with smaller size. The aerosol number concentration, however, is dominated by particles with $d_p < 100$ nm which are not activated to cloud droplets as efficiently as particles with larger size, and thus not removed when the cloud precipitates. Hence the in-cloud scavenging affects the removal much less when the total particle number is inspected than in the case of large particles dominating the mass loading.





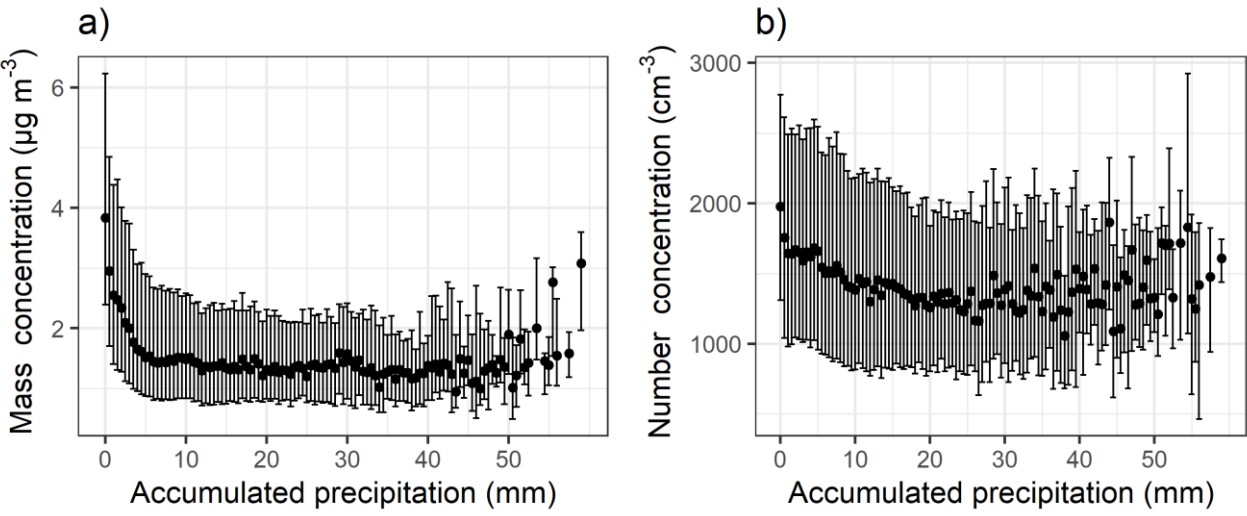

**325**

**Figure 1** Total particle ($d_p$ = 3-1000 nm) mass (a) and number (b) concentration as a function of 0-50 mm accumulated precipitation along the 96-hour HYSPLIT airmass trajectories. The black dots show the median values and bars highlight the 25th-75th percentiles for each 0.5 mm bin of accumulated precipitation. The figure includes DMPS data between January 2005 and August 2019.

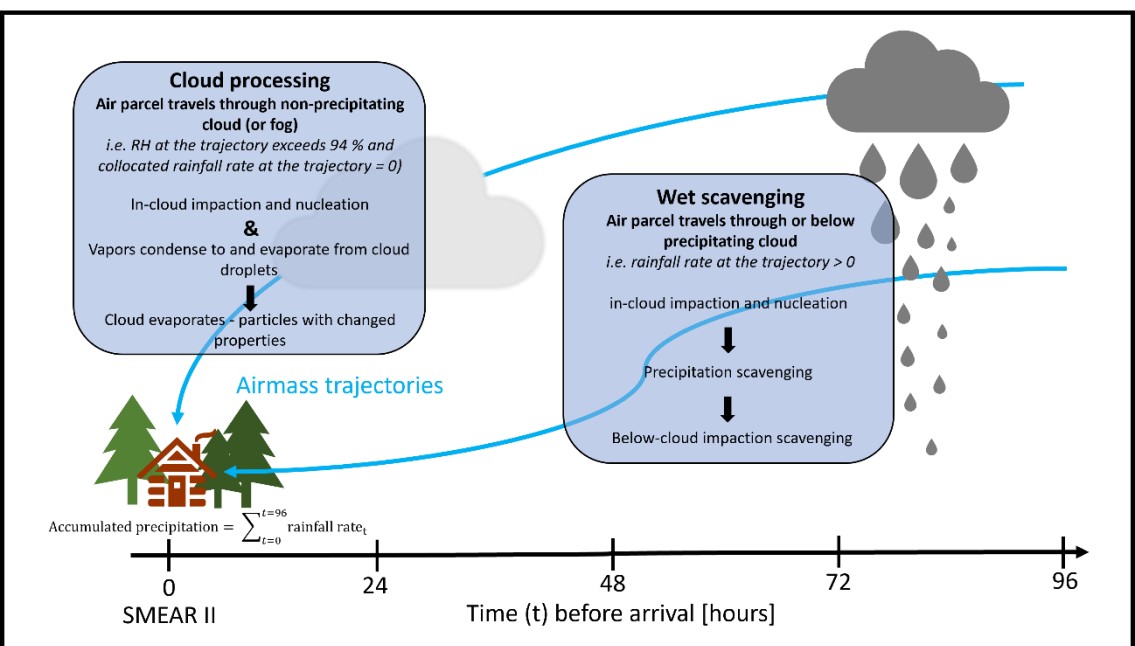

**330**  **Figure 2** Schematic visualizing the wet processes along airmass trajectories in the Lagrangian framework. Travelling particles experience different conditions en-route thus alternating the observed particle concentrations (through scavenging) and composition (through cloud processing) at the SMEAR II.



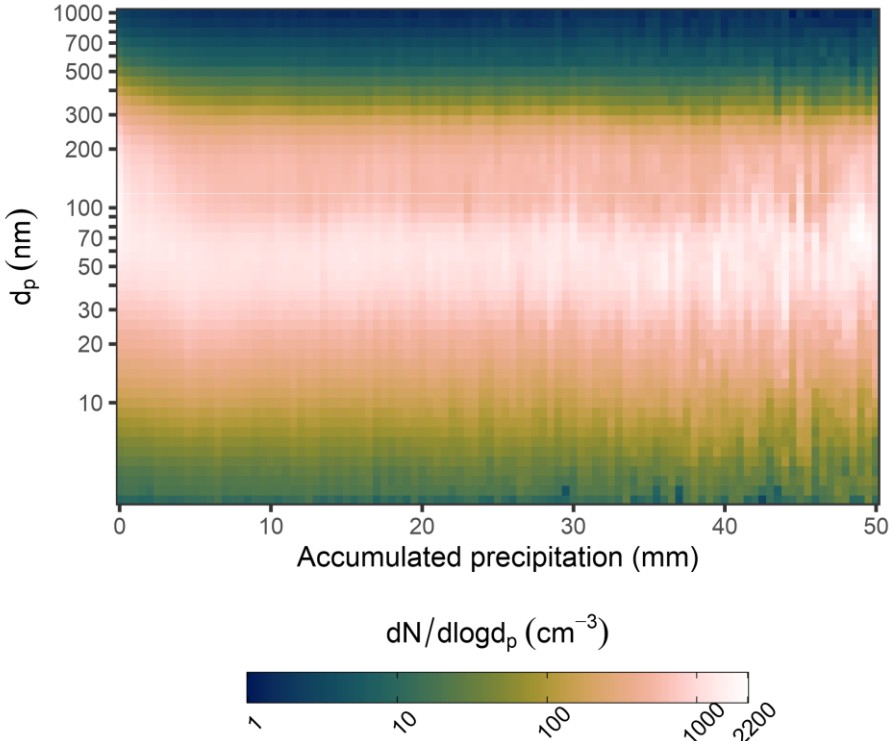

**Figure 3 The aerosol size distribution ($d_p$ = 3-1000 nm) as a function of the 0-50 mm of accumulated precipitation. Data is shown as**
**medians for binned accumulated precipitation (bin size 0.5 mm). The figure includes DMPS data between January 2005 and August**
**2019.**

**3.2 Effect of wet scavenging on the aerosol composition**

The effect of accumulated precipitation on the different chemical components (organics, sulfate, black carbon, nitrate,
ammonium and chloride hereafter Org, $SO_4$, eBC, $NO_3$, $NH_4$ and Chl, respectively) can be investigated using the long-term
observational data (see Section 2.1 for details). In this study, our focus is on $SO_4$, Org and eBC and the other chemical
species obtained with ACSM are included in the supplementary material as their mass concentrations are generally relatively
low at SMEAR II (Heikkinen et al., 2020). To investigate possible seasonal differences in the wet scavenging of the
particles, we divided the data based on monthly median temperatures ($T_m$). Months which have $T_m < 10$ °C (calculated from
data between 2005-2019) include January, February, March, April, October, November, and December and hereafter
referred as "cold" months. Months that have $T_m > 10$ °C include May, June, July, August, and September and are referred as
"warm" months. This division into two seasons is used to ensure enough data points for each bin, as the chemical
composition measurements are more limited than the particle size distributions. Each of the studied chemical component





shows exponential decrease as a function of accumulated precipitation (Figure 4a-c), and similar decreases is also seen if the reanalysis data is changed (Figure S22a-c).

To investigate the possible differences in the removal efficiency for different species, we normalized the median mass concentration values with the median mass concentration value when the accumulated precipitation is zero (Figure 4d-e). The median mass concentrations (and 25th-75th percentiles) for non-precipitating trajectories for Org, eBC and $SO_4$ were 3.77 (2.18-5.49), 0.28 (0.17-0.45) and 0.52 (0.34-0.88) µg m$^{-3}$ for warm months and 2.03 (1.22-3.37), 0.48 (0.26-0.85) and 1.01 (0.51-1.53) µg m$^{-3}$ for cold months, respectively. Org mass concentration, for example, is much higher during warm months

due to strong local biogenic activity whereas $SO_4$ mass concentration in warm months is ~50 % of that in cold months, suggesting the two seasons introduced here capture the typical seasonal characteristics in this region reasonably well. $SO_4$ is removed less efficiently than Org and eBC during warmer months during the arrival of the airmasses to SMEAR II, as can be seen from Figure 4d. During the first 10 mm of accumulated precipitation, the normalized particle mass has decreases from 1 to 0.62 for $SO_4$, whereas Org and eBC have reached 0.37 and 0.32, respectively. This could indicate that more of the

$SO_4$, compared to Org and eBC, is distributed to smaller particles during warmer months which reduces both CCN activation potential and thus removal of activated particles by rainfall. The observed differences cannot be explained by below-cloud scavenging, as the composition measurements are representative for particle sizes above 70 nm (up to 1 µm), which are not efficiently scavenged below-cloud (e.g. Croft et al., 2009). Also, stronger contribution of local sources of $SO_4$ during warm months could result in lower removal efficiency.

Conversely, during the colder months (Figure 4e), $SO_4$ is removed slightly more efficiently than Org and BC (decreases from 1 to 0.39, 0.34 and 0.28 for Org, eBC and $SO_4$, respectively, with the first 10 mm of accumulated precipitation). The differences of the removal efficiency between the investigated components are smaller during colder months when compared to warmer months, suggesting the species are internally mixed during colder months (as $SO_4$ and eBC, for example, have very different hygroscopicity, but still are removed almost as efficiently). The trajectories derived with the GDAS

meteorology precipitate on average less (Figure S19) than those derived with the ERA-Interim meteorology (Figure S1). This explains the less efficient removal of the standardized particle masses in Figure S22d-e. It is also possible that the seasonal differences in cloud types and cloud cover fractions within one grid box in the reanalysis dataset could have an effect to the observed differences between the wet scavenging efficiencies. The relative contribution of wet scavenging is 5-10 times smaller during the warmer months (Tables S4-S6), which show less defined removal by accumulated precipitation

compared to warmer months. Regression coefficients indicate more efficient removal during colder months for all species. Comparing Figure 4d and e, we see that the data points are much more scattered during the warmer months for all three species. This could indicate a larger contribution from local production and thus we can observe relatively large mass concentrations in SMEAR II even with high accumulated precipitation values along the airmass route. Figure S8 shows the particle number size distribution for $d_p$ = 50-700 nm (electrical mobility diameter), roughly representative for the sizes

measured by ACSM (ca. 75-1000 nm in vacuum aerodynamic diameter) as a function of the accumulated precipitation (like Figure 3) for the two temperature regimes. We clearly observe a relatively high number of particles, especially smaller ones,





during warmer months despite the high values of accumulated precipitation along the trajectory. The decreases seen in the number size distribution for the different particle sizes during the first 10 mm of precipitation are steeper during the colder months. Similar behaviour (steeper decrease during colder months) is observed for $SO_4$ mass concentration in Figure 4e.

Based on the statistical modelling, the contribution of local meteorology to the organic mass concentration, for example, is an order of magnitude larger during warmer months (group 3 in Table S4). For $SO_4$ and eBC, large difference between the seasons is seen in terms of long-range transport (group 5 in Table S5 and S6). Long-range transport has relatively small contribution (Section 2.3.2) in the mixed effects models during the warm months compared to colder months (i.e. the variable group is less crucial for the model with data from warmer months), and as the wet scavenging discussed here takes

place along the airmass route, defined removal is not observed (as seen in Figure 4d). Conversely, during cold months the relative contribution of long-range transport (and wet scavenging, group 4a) for $SO_2$ is much larger, thus we see more defined removal (i.e. less scattering of the data points) during the colder months in Figure 4e for eBC and $SO_4$.

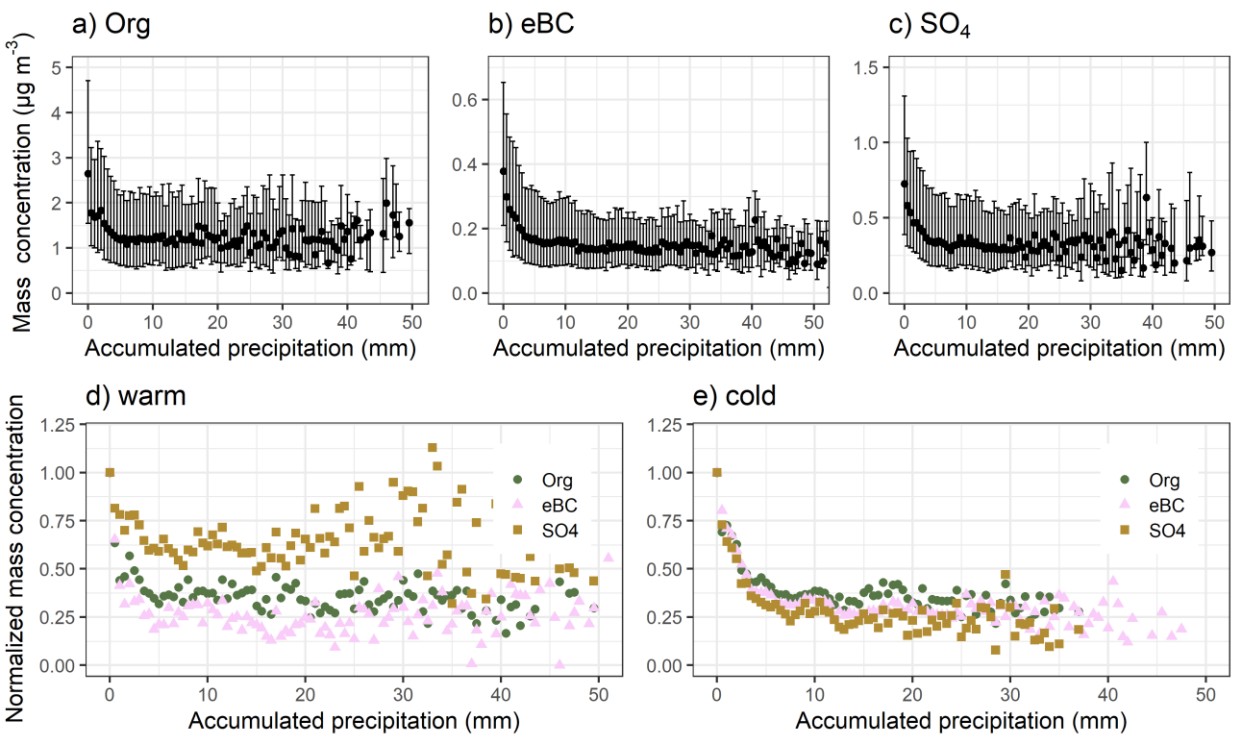

**Figure 4 Particle mass concentration of a) Org, b) eBC and c) $SO_4$ as a function of accumulated precipitation along the 96-hour HYSPLIT airmass trajectories. The black dots in the top row show the median values and bars highlight the 25th-75th percentiles for each 0.5 mm bin of accumulated precipitation. Bottom row shows normalized particle masses (calculated from the medians) with temperature separation. Medians and normalized medians are shown for each bin having 10 or more data points. The figure includes data between 2006-2019 for eBC and 2012-2019 for Org and $SO_4$.**



### 3.3 Effect of in-cloud processing on aerosol concentrations and composition

To investigate the possible effects of in-cloud processing on aerosol composition, we took advantage of the relative humidity (RH) provided by the HYSPLIT model along the airmass trajectories. We selected a limit of RH > 94 % (similar to Tunved et al., 2004) to identify cases that we assume the airmass is inside a cloud (or fog, as we do not separate these cases) as visualized in Figure 2. We would like to note that even if the approximation for the in-cloud cases is not very accurate based on the RH values only, the humidity in these cases is high enough for the particles to have taken up significant amounts of water. Strong hygroscopic growth can be observed before activation, and e.g. for inorganic salts the deliquescence RH is well below 94 % (e.g. Cruz and Pandis, 2000; Zieger et al., 2017; Lei et al., 2018). Thus, it is safe to assume the aqueous phase processes, whether in cloud or inside figs, are taking place when RH of 94 % is exceeded.

The observations were divided into 3 groups based on the conditions (precipitation and clouds) the arriving airmasses have experienced during the last 24 hours to investigate if precipitation and in-cloud aqueous phase processing affect the particles differently. Group 1 represents the cases where the arriving airmasses have not experienced precipitation or clouds (i.e. RH < 94 %) within the last 24 hours before arriving at SMEAR II. Group 2 represents cases where airmasses have experienced precipitation within the last 24 hours. Group 3 represents cases where the airmass has experienced RH > 94 % (i.e. in-cloud conditions) but no precipitation within the last 24 hours. These definitions are summarized in Table 2. We restrict trajectories to the 24 hours prior to arrival to ensure enough observations corresponding to the trajectories in each group, especially in Group 3 which has the strictest criteria.

Figure 5 shows the median mass (a) and number (b) concentration of the accumulation mode ($d_p$ = 100-1000 nm) particles based on their experiences of precipitation and high humidity conditions (RH > 94 %) during the last day before arrival to SMEAR II, as described in Table 2. Figure 5b shows that the accumulation mode number concentration is lower if the airmass has experienced precipitation (group 2) or high humidity conditions (i.e. clouds) without precipitation (group 3), compared to the case when the airmass has not experienced precipitation or in-cloud conditions (group 1). When the mass concentration (Figure 5a) is investigated, we see higher mass concentration for group 3 compared to group 1, suggesting that in non-precipitating, high RH conditions the aerosol mass increases due to aqueous phase processes. Identical observations can be made if the GDAS reanalysis meteorology is used in the calculation of the trajectories (Figure S23).

To investigate further the observed increase in accumulation mode mass concentration (Figure 5a) when the airmass had experienced high humidity conditions, we investigated the bulk aerosol composition. We focus on group 1 (no wet processing at all) and group 3 (high humidity conditions, but no precipitation) as we are now interested in the increase in mass concentration between those groups, as shown in Figure 5a. Here, we concentrate on the mass concentrations of organics (Org) and sulfate ($SO_4$), and black carbon (eBC) measurement data. Figure 6 shows the median particle mass concentrations (see Fig. S9 for mass fractions) for each of these chemical species for clean and more polluted airmasses with the temperature division for the wet processing groups 1 and 3. The division of the trajectories into "clean" and "polluted" sectors was made by assigning any trajectories that visited latitudes below 60 degrees North into polluted sector, and rest to





the clean. Thus, our final subgroups are WC (warm, clean), WP (warm, polluted), CC (cold, clean) and CP (cold, polluted). This approach was applied to make sure the observed changes in the concentration of species are indeed related to the

aqueous phase processing and to exclude the influence of artefacts arising from possible association of different source areas to different meteorological conditions (i.e. group 1 vs group 3). Further justification for our choice of these sectors can be found in the chapter below. Increases exceeding the error limits in $SO_4$ concentration (Figure 6) are observed between wet processing groups 1 and 3, suggesting $SO_4$ formation in the aqueous phase, while no significant changes are observed for Org and eBC, except for CP subgroup (Figure 6d) where Org shows a decrease. The patterns in the mass concentrations for

each species between the groups 1 and 3 showed similar behaviour when we increase the time (0-24 h into e.g. 0-36 h or 0-48 h) used to determine the classes.

**Table 2  Definitions for the wet processing groups. Availability shows the percentage of trajectories relative to total number of trajectories belonging to the wet processing groups.**

| Group | History during the last 0-24 hours before arrival to SMEAR II | Quick summary | Availability (%) |
|---|---|---|---|
| 1 | Airmass has not experienced precipitation or RH > 94% | No precipitation or in-cloud processing | 24.5 |
| 2 | Airmass has experienced precipitation | Wet scavenging | 62.2 |
| 3 | Airmass has experienced RH > 94 % but not precipitation | Only non-precipitating clouds (in-cloud processing) | 13.3 |

*Definitions are according to description of explained quantities in Sect. 2.2





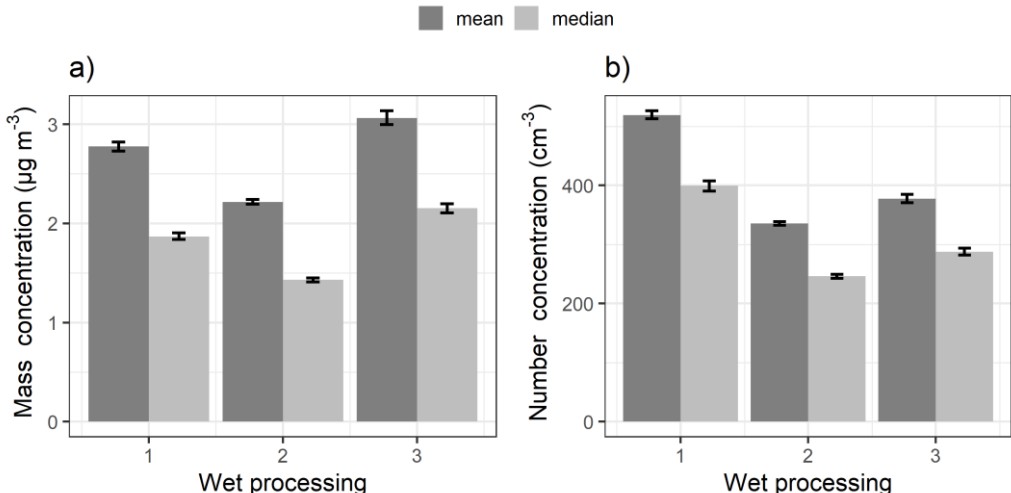

**Figure 5 Median and mean accumulation mode ($d_p$ = 100-1000 nm) particle mass (a) and number (b) concentration for wet processing groups described in Table 2. The figure includes DMPS data between January 2005 and August 2019. The whiskers show the 99% confidence intervals from 1000 bootstrap replicates.**

Two sectors were used to distinguish the more polluted and mostly clean airmasses, as more detailed division on air mass source areas is not possible because of the limited amount of data available especially for the group 3 cases. Even though the division is relatively rough, it does separate the airmasses quite well, especially as we have already excluded the highly polluted airmasses arriving from the Kola peninsula and emissions arriving from the nearby sawmills. For example, in a former study from Kulmala et al. (2000), trajectories arriving from the Arctic ocean coincide with a low number of

accumulation mode particles and low $SO_2$ concentration and, therefore, airmasses arriving from that sector are classified as "clean". Sogacheva et al. (2005) also presented similar sources for accumulation mode particles. The same classification is used in this study. Sectors from Kulmala et al. (2000) including the southeast of Russia and Central Europe showed highest accumulation mode number and $SO_2$ concentrations and are classified as "polluted" also in our study. We selected two sectors to maintain high enough statistics for the composition measurements to achieve reliable results. In addition, the

temperature-based division, discussed in section 3.1, gives additional insight when separating the airmasses. Riuttanen et al. (2013), conducted trajectory analysis to investigate trace gases observed in SMEAR II, and our temperature-based division coincides well with the seasonality of $SO_2$ concentration.

To further confirm that the seasonal patterns of trace gases like $SO_2$ and aerosols shown in Riuttanen et al. (2013) also hold for our study period, source contribution analysis was conducted. No major changes in the source areas are observed (Figure

S15 and Figure S16 as examples for accumulation mode particle number concentration and $SO_2$ mixing ratio, respectively) when compared to the study from Riuttanen et al. (2013). We can observe a clear difference in the total mass concentration between clean and more polluted airmasses in Figure 6, indicating our sector division into mainly clean and more polluted is suitable. The average particle size distribution with these sector- and temperature-based divisions is shown in Figure S10,





but reader should be aware that the composition measurements do not represent particles with $d_p < 70$ nm. There is no clear

difference in the geographical distribution of air mass trajectories between the wet processing groups 1 and 3 shown in

Figure 7, thus we can conclude that the observed differences in $SO_4$ are not associated to different source areas of airmasses

between the groups 1 and 3. Group 3 includes less trajectories due to our strict definitions of the airmass history groups, but

the trajectories are arriving from similar areas in both group 1 and group 3. Further, to exclude possible influence caused by

transport of $SO_4$ from the oceans into our results (e.g. dimethyl sulfide derived sulfate aerosols, which could show up as high

$SO_4$ concentrations coinciding with high humidity conditions if the airmass trajectory passes close to the sea surface, e.g.

Barnes et al., 2006), we investigated the vertical transport of the airmasses. This analysis showed no evidence (see Section

S4) that this type of transport is significantly influencing the results presented here.

Thus, based on the airmass history analysis presented above and conclusions drawn regarding $SO_4$ transport from oceans, we

can state that the observed increase in $SO_4$ is likely due to aqueous phase chemistry, where $SO_2$ is oxidized in the aqueous

phase to form $SO_4$ (e.g. Barth et al., 2000; Ervens, 2015; McVay and Ervens, 2017). A relative increase of 45-63 % is

observed between groups 1 and 3 in Figure 6b-d (airmass histories WP, CC, CP), and the largest increase is observed for

more polluted airmasses during colder months (CP). The increase in $SO_4$ concentration is not seen for clean airmasses during

the warmer months (Figure 6a, WP) as e.g. $SO_2$ concentration, an important precursor for $SO_4$ formation in-cloud, are lower

in cases for airmasses coming from northern areas and for the warm season (Kulmala et al., 2000; Riuttanen et al., 2013). For

the colder months and more polluted airmasses (Figure 6b-d), the increase in $SO_4$ is more pronounced due to more precursor

$SO_2$ available for $SO_4$ production in-cloud (e.g. Paulot et al., 2017). The increasing trends in $SO_4$ mass concentration and

mass fraction are similar to what is shown in Figure 6, when all chemical species measured by the ACSM are considered

(mass concentrations in Figure S11, mass fractions in Figure S12). In addition, small increases in the mass concentration of

$NH_4$ can be observed for group 3. This is likely because of the enhanced uptake of ammonia from the gas phase with

increasing sulfate fraction (Harris et al., 2014). Changes in the $SO_4$ concentration due to aqueous phase processing are

similar also when the GDAS reanalysis meteorology is used. In-cloud formation of $SO_4$ is also supported by the statistical

model in which we consider the other factors also affecting the local particle concentrations (Table S8).

The mass concentration of Org shows no difference when comparing case 1 to case 3 in Figure 6a-c but shows a decrease in

the cold and polluted subgroup (Figure 6d, CP, relative decrease 32 %). With trajectories using GDAS reanalysis

meteorology, decrease in Org exceeding the error limits is also seen, but for the warm and clean airmasses (Figure S24a,

WC). Previous studies have shown an increase of organic mass through aqueous phase production of SOA (Blando and

Turpin, 2000; Ervens and Volkamer, 2010; Ervens et al., 2018). For example, Ervens et al. (2018) investigated the formation

of aqSOA with parcel models from gas-phase precursors of toluene, xylene, and ethylene. In SMEAR II, the gas phase

precursors from biogenic sources are dominated by monoterpenes specially during warm months (e.g. Hakola et al., 2012;

Patokoski et al., 2015; Barreira et al., 2017; Heikkinen et al., 2021).

During the colder months (Figure 6c-d), the airmasses are likely to have more anthropogenic influences and thus a different

VOC profile (e.g. Patokoski et al., 2015), but the formation of aqSOA is still negligible when the total Org mass is



investigated in this area of northern Europe. It has also been suggested that water soluble SOA (originating from other sources than aqueous phase processing) in the cloud droplets can become oxidized and become more volatile leading to

evaporation. This could lead to a decrease in total SOA mass, even though the aqueous phase SOA mass is formed (Ervens et al., 2018). In addition, the increase in $SO_4$ can increase the acidity of the droplets which might increase the evaporation of organic acids leading to a decrease in the organic mass (Ervens et al., 2018). Our data suggests that the local photochemically driven SOA production at SMEAR II (and surrounding areas) dominates over the aqueous phase SOA formation especially during the warm months. This is supported by the solar radiation values measured at SMEAR II, as they

are much lower for group 3 than for group 1 for all cases. Hence, decreased SOA formation due to the decreased photochemical activity could compensate for the in-cloud aqueous phase SOA formation resulting in comparable total organic mass when groups 1 and 3 are compared. Our results indicate, that in the boreal forest dominated Northern Europe the in-cloud aqueous phase SOA production has negligible impact on total observed organic mass both during warmer and colder seasons, and in the case of clean and polluted airmasses. The Org mass concentration also shows no increase due to

clouds when the other factors affecting the local concentrations are considered with the mixed effects model (Table S7). When investigating the composition of the particles as a function of time in RH > 94 % (Figure 8) we observe an increase in sulfate mass fraction with time spent under the high humidity conditions, especially for the more polluted airmasses which also have more $SO_2$ available for the in-cloud production of $SO_4$ (Figure 8a). This trend is not seen when looking at the time the airmass was influenced by precipitation (Figure 8b) indicating precipitation acts mainly as a sink for the particles,

whereas high humidity conditions, i.e. in-cloud aqueous phase processing, also alters the particle chemical composition. Again, the increasing trend in $SO_4$ fraction when the airmasses arrive from cleaner areas is more subtle (Figure 8c). Same trends are observed when all species from the composition measurements are investigated (Figure S13). These results suggest that not only the experience of in-cloud aqueous phase processing (Figure 6) affects the particle composition, but also the time spent in cloud has an effect. Unfortunately, with this type of analysis of the time dimension, we were not able

to apply the temperature-based division in addition to the sector division as it would limit the number of observations for the long exposure times too much to obtain statistically reliable results. Results obtained with the GDAS reanalysis meteorology (Figure S26) agree well with those from Figure 8. The very long exposure times (> 60 hours) of precipitation are missing from the GDAS derived trajectories due to lower occurrence of precipitation events compared to the ERA-Interim derived trajectories (see Figures S1 and S19).

To investigate which particle sizes are most affected by the increasing mass of $SO_4$, the DMPS size distribution was divided into 7 classes with particle dry diameter ranges (in nm) of [3, 30], (30, 50], (50, 100], (100, 200], (200,350], (350, 600] and (600, 1000]. Using the airmass history groups presented in Table 2, the mass concentrations for these size classes are shown in Figure 9 corresponding to the sector and temperature divisions first shown in Figure 6. The mass concentration is larger if the airmass has experienced high humidity conditions (group 3) for particle with diameters between 200-1000 nm, when

compared to group 1 where there have not been any wet processes in the last 24 hours. Same observation can be made with the trajectories using GDAS meteorology (Figure S27). The increase is clearest size ranges (200, 350] and (350, 600] and the





size range (600, 1000] shows a very minor increase in mass for some subgroups. The same changes are also seen when the

particle mass data is strictly limited to simultaneous observations with the composition (Org, $SO_4$, BC) measurements

(Figure S14). These results suggest that the $SO_4$ formed via in-cloud aqueous phase processes is mainly distributed to

particles having a dry diameter between 200 - 1000 nm.

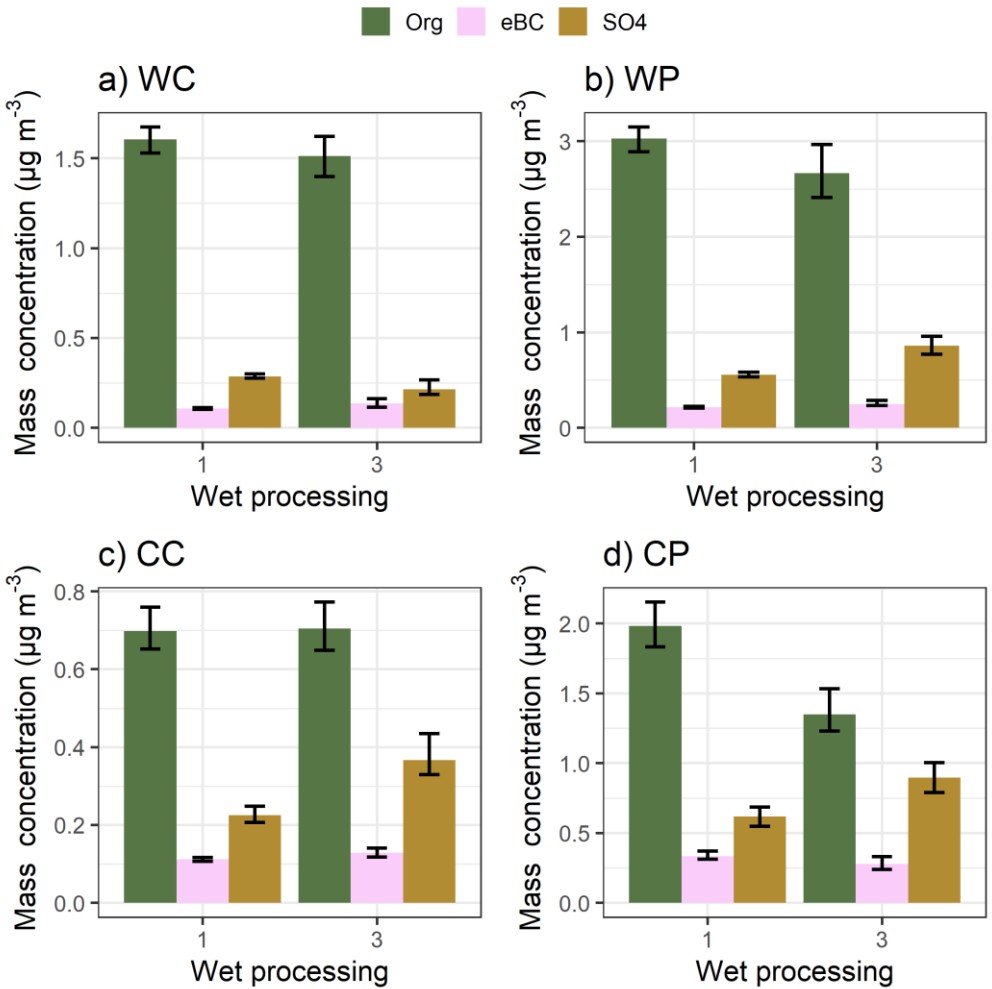

**Figure 6 Median particle mass concentration for Org, eBC and SO₄ for wet processing groups 1 and 3 as described in Table 2. Subplots show the airmass sectors (clean and polluted) with the seasonal (warm and cold) division followingly: a) warm and clean, b) warm and polluted, c) cold and clean and b) cold and polluted. The figure is based on simultaneous observations of these three species between March 2012 and August 2019. The whiskers show the 99 % confidence intervals from 1000 bootstrap replicates. Note the different y-axis limits in each subplot.**





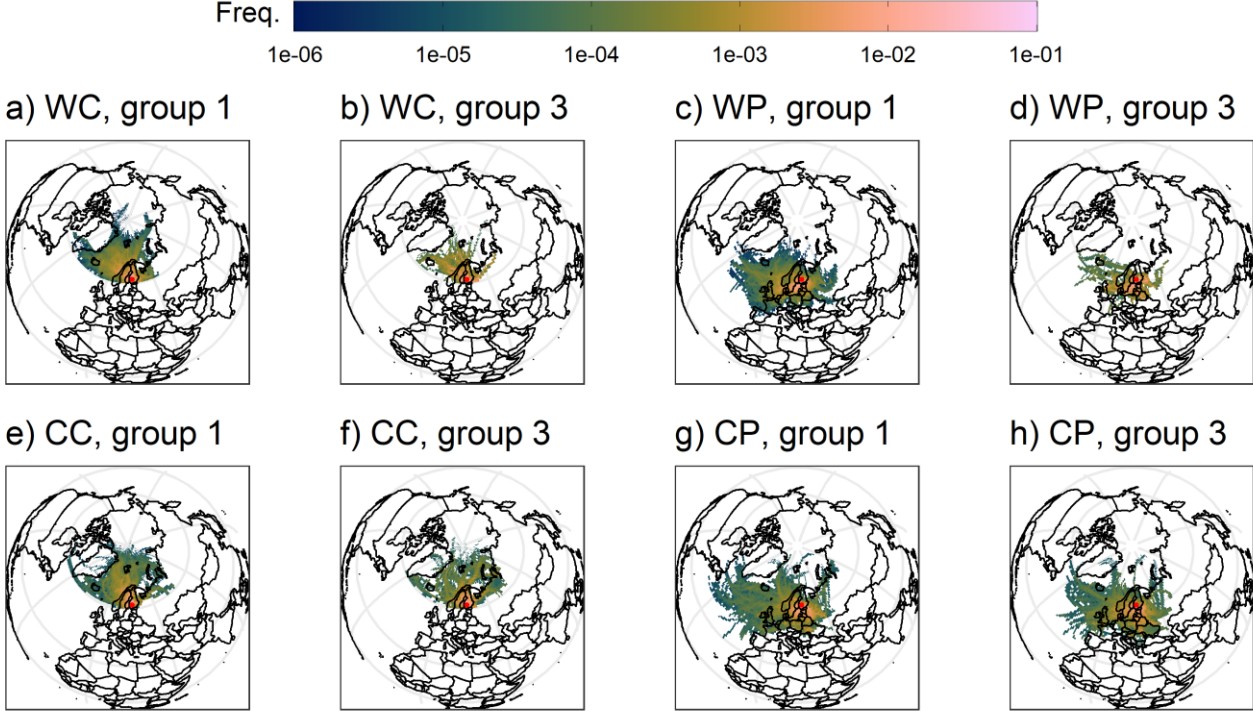

**Figure 7 96-hour airmass history for the wet processing groups (1 and 3) with the sector (clean and polluted) and temperature (warm and cold) division. Subplots show a)-b) warm and clean, c)-d) warm and polluted, e)-f) cold and clean and g)-h) cold and polluted. Colour scale shows the frequency (crossings in each 1° x 1° grid point divided by total number of crossings in each group) of trajectories crossing a grid point. The groups 1 and 3 correspond to the airmass history groups presented in Table 2. The red dot shows the location for SMEAR II.**

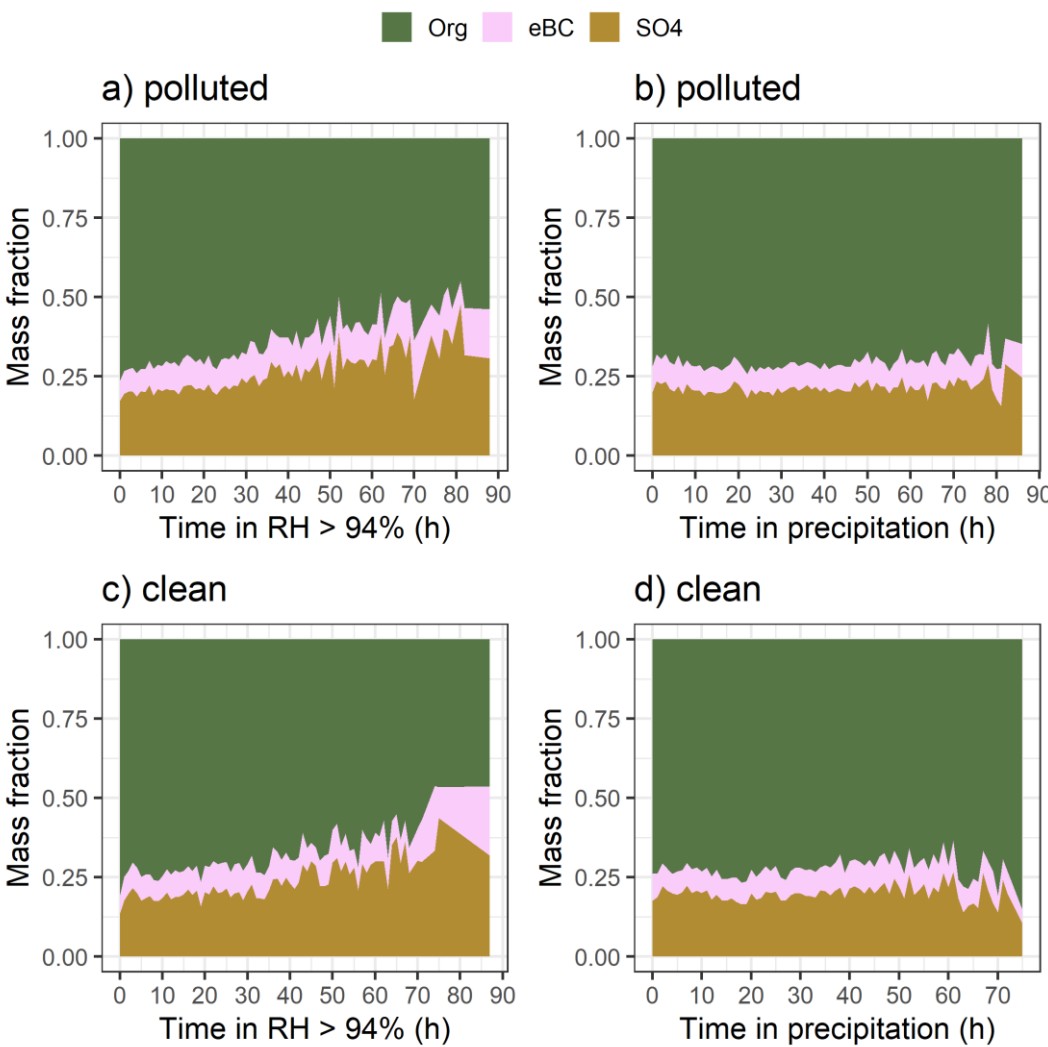

**Figure 8** The mass fractions of Org, SO₄ and eBC for clean and more polluted airmasses as a function of time spent in RH > 94 % (a and c) and in precipitation (b and d): Figure shows median values for each 1-hour bin, if 10 or more data points were available in the bin. The figure is based on observations between March 2012 and August 2019.



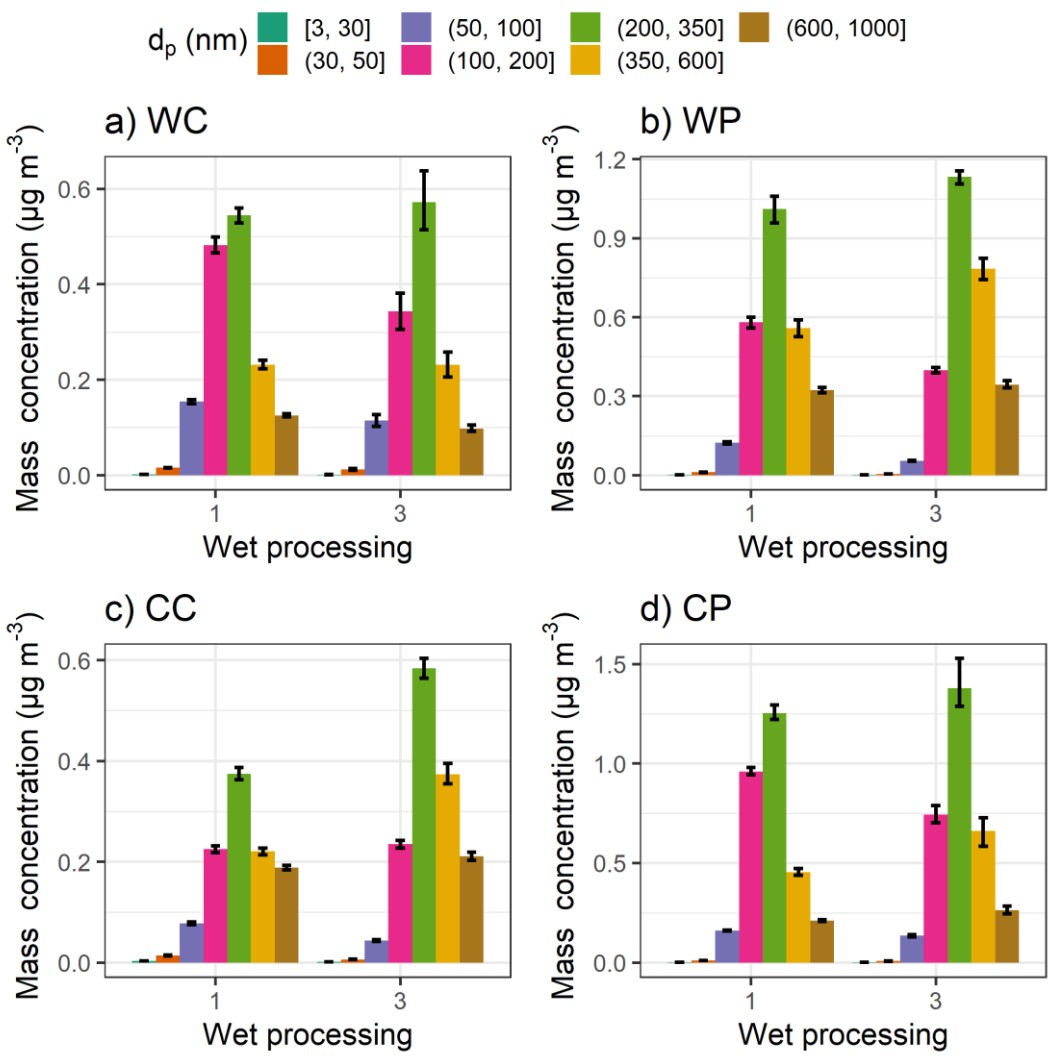

**Figure 9 Median particle mass concentration for size bins derived from the DMPS measurements for wet processing groups 1 and 3 as described in Table 2. Subplots show the airmass sectors (clean and polluted) with the seasonal (warm and cold) division followingly: a) warm and clean, b) warm and polluted, c) cold and clean and b) cold and polluted. The figure includes all available data between January 2005 and August 2019. The whiskers show the 99 % confidence intervals from 1000 bootstrap replicates. Note the different y-axis scale in each subplot.**







## 4 Conclusions

In this study we investigated how wet processes taking place in clouds including aerosol wet scavenging and aqueous phase oxidation along airmass trajectories affect to the observed sub-micron aerosol population in SMEAR II station, Hyytiälä, Finland which represents the boreal environment. Our first objective was to investigate how efficiently different chemical species are removed from the atmosphere by precipitation. Based on over a decade long timeseries, we observed an exponential decrease in particle mass as a function of accumulated precipitation along the trajectory, similar to what has been reported in earlier studies. We conclude that the in-cloud wet scavenging dominates over below-cloud precipitation scavenging, especially for the accumulation mode (100 nm $< d_p <$ 1 µm) sized particles. Scavenging of particle mass was more effective during colder months especially for sulfate aerosol, whereas the other investigated aerosol chemical species behaviour was more alike. Statistical mixed effects models also showed removal by accumulated precipitation for all species, suggesting more efficient removal in colder months. In addition, the scavenging efficiencies were relatively similar between the species in colder months, suggesting the particles are internally mixed and the different species are distributed to similarly sized particles. During warmer months, it is likely that strong local particle production in SMEAR II effectively masks the wet scavenging along the trajectory, thus relatively high particle mass concentrations are observed despite high values of accumulated precipitation. This was supported by the statistical modelling, in which e.g. the relative contribution of local meteorology on organic aerosol production was much larger during the warmer months. Seasonal differences in cloud types and cloud cover fractions within one grid box in the reanalysis dataset may also have an effect to the observed differences in wet scavenging efficiencies between the seasons.

Our second objective was to investigate how aqueous phase processing taking place in clouds alters the particle mass concentration and composition. Our study reveals a significant in-cloud formation of sulfate mass, but aqueous phase SOA formation could not be identified by the analysis. In-cloud processing was separated by using relative humidity as a proxy for the airmass to have experienced clouds and the precipitation data along the airmasses was used to separate non-precipitating clouds. An increase in accumulation mode particle mass was observed for airmasses that had recently been in-cloud when compared to clear sky airmasses (airmasses with no experience of clouds or rain during the last 24 hours). The chemical composition of accumulation mode particles was studied and the observed increase in particle mass can be mostly attributed to in-cloud production of $SO_4$. Our analysis shows that sulfate mass concentrations increased 45-63 %, depending on season and airmass origin, due to in-cloud sulfate formation. Furthermore, the increase in sulfate mass fraction was higher when the air mass had spent more time in high humidity conditions. Statistical mixed effects model, in which additional factors affecting local $SO_4$ concentrations are considered, also supported in-cloud $SO_4$ formation whereas no formation of Org or eBC mass was observed.

Airmass history analysis was applied to separate airmasses originating from different sources (more polluted and mostly clean) in addition to the temperature-based seasonal division to investigate how different conditions along the airmass affect the observed increase in $SO_4$ mass concentration due to in-cloud processes. When airmasses originated from areas with more





pollution sources producing gaseous $SO_2$, a greater increase in $SO_4$ mass was observed due to more SO2 being available in

the gas phase to be oxidized in-cloud to form $SO_4$. Aqueous phase production of organic mass was not observed during the warm months since monoterpenes dominate both the biogenic VOCs and total VOCs in this region. In wintertime anthropogenic emissions dominate over biogenic emissions, thus aqueous phase production of organic mass was not observed during the cold months either. We were also interested if the effects of aqueous phase processes are different for particles of different size. Therefore, changes in the particle size distribution were investigated to determine into which

particle sizes the observed mass increase in $SO_4$ is mostly distributed. Increases in particle mass were observed for sizes larger than 200 nm, whereas smaller sizes showed a decrease in some cases.

Finally we also compared the trajectories based on different reanalysis meteorologies (ERA-Interim and GDAS) as an additional robustness test for our results. Both approaches gave nearly identical results and thus the same conclusions could be drawn. Trajectories obtained with the GDAS reanalysis meteorology had in general less precipitation events, and thus the

scavenging efficiency of the investigated species was lower compared to the results obtained with ERA-Interim based trajectories. $SO_4$ formation in the aqueous phase was observed to significantly contribute to the total $SO_4$ mass with both approaches whereas formation of aqSOA was not detected. Precipitation values derived from the trajectory model at SMEAR II agreed also well with the locally measured precipitation. The results from this study offer an interesting insight in using air mass history analysis to study aerosol-cloud interactions and facilitates the comparison of observed aerosol wet

scavenging and in-cloud processing with outcomes of larger scale models. This study highlights that the ability of global models to simulate aerosol composition and size distribution, especially away from source regions, can be improved by improving the description of size dependent wet removal of different aerosol compounds. Our analysis also provides a good platform for evaluating the ability of models to simulate in-cloud chemical formation of aerosol. Further analysis is needed to investigate in more detailed the effect of clouds and precipitation on aerosol dynamics and detailed changes in size

distributions.




**Data availability**

Raw data were collected by INAR, University of Helsinki. Field data (particle size distributions, meteorological variables, black carbon, and trace gases) are available from https://smear.avaa.csc.fi/download. The ACSM data on aerosol composition is available at EBAS data base at http://ebas.nilu.no/. The pre-processed HYSPLIT trajectory data can be

obtained from the corresponding author and the trajectories can be freely calculated at the webpage https://www.ready.noaa.gov/HYSPLIT_traj.php.

**Author contribution**

A.V proposed the study and designed the research questions. S.I had the lead role in data analysis with supporting

contribution from P.K and D.P. Results were interpreted by S.I, A.V, P.K, D.P, S.M, T.Y and L.H. The manuscript was written by S.I. with supporting contributions from P.K, S.M, D.P and A.V. All co-authors commented and edited the manuscript. L.H. performed the ACSM measurements and data processing. K.L. performed the aethalometer measurements and data processing.

**Competing interests**

The authors declare no competing interest.

**Acknowledgements**

We thank technical and scientific staff in SMEAR II station. Tuomo Nieminen is gratefully acknowledged from his

contribution during initial trajectory analysis.

**Funding information**

Financial support from the European Union's Horizon 2020 research and innovation programme (project FORCeS grant No. 821205, project COALA grant No. 638703), European Research Council (Consolidator grant INTERGRATE No. 865799)

and Knut and Alice Wallenberg foundation (Wallenberg Academy Fellowship project AtmoRemove No 2015.0162) is gratefully acknowledged. This research has also been supported by the Academy of Finland (grant No. 317373 and 317390), Academy of Finland Flagship funding (grant No. 337550) and the Academy of Finland competitive funding to strengthen university research profiles (PROFI) for the University of Eastern Finland (grant No. 325022). The work of S.I was financially supported by the University of Eastern Finland Doctoral Program in Environmental Physics, Health and Biology.



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
