# Peer review of "The effect of clouds and precipitation on the aerosol concentrations and composition in a boreal forest environment"

_Atmospheric Chemistry and Physics, 2022_

## Author Comment (AC1)

**Response to reviewers**

**Bolded text indicates the comments from Reviewers,** normal text shows our response *and italic changes made to the manuscript*.

**REVIEWER #1**
**General scientific comment:**

**The study offers interesting findings to improve the understanding of wet removal processes of aerosols. In addition, the phenomenon of cloud processing is investigated for the case that wet removal has not taken place. The analysis focusses on the changes of physical parameters as e.g. particle number as well as on the change of the chemical composition of the aerosol population. In general, I have only few comments that I like to be addressed before final publication.**

We thank the Reviewer for the positive feedback, comments and suggestions which have improved our manuscript. Please see our responses and changes made below from our point-by-point answers.

**Detailed scientific comments:**

**Abstract**

**Line 22:**

**… particle number size distribution …**

This is now modified as suggested.

**Line 27:**

**… as a function of …**

 Modified as suggested.

**Line 34:**

**… that had no contacts to clouds or …**

The air parcel trajectories we have from HYSPLIT simulate the large-scale airmass transport. As they are derived from the reanalysis data (which has a resolution of ~1 degree corresponding around 100 km x 100 km, see e.g. Stengel et al., 2018), they do not resolve any sub-grid scale processes and will not capture transport through individual clouds (which could be in the order of hundreds of meters). The airmass transport routes (and the "clouds" in our study) from HYSPLIT can thus be tied to the average meteorological properties of reanalysis grid box that the trajectories cross. Therefore, stating "in contact with clouds" could be misleading. We have now reworded the referred sentence and added more explanation of the clouds in our study to the methods section.

Modifications to abstract:

*We compared airmasses with no experience of approximated in-cloud conditions or precipitation during the past 24 hours to airmasses recently inside non-precipitating clouds, before they entered SMEAR II.*

Added sentences to methods:

*The air parcel trajectories we have obtained from HYSPLIT simulate the large-scale airmass transport. As trajectories are derived from the reanalysis data with 1 degree resolution (~100 km x 100 km), they do not resolve any sub-grid scale processes. Therefore, they will not capture transport through individual clouds, which could be in the order of hundreds of meters. The airmass transport routes, and the clouds/precipitation in our study, can thus be tied to the average meteorological properties of reanalysis grid box that the trajectories cross.*

**Line 27:**

**… particle number size distribution measurements …**

We assume the Reviewer here referred to line 37, which is now modified as suggested.

**Introduction**

**Line 45:**

**… particle number size distributions …**

We went through the whole manuscript and modified all instances of particle size distribution into particle number size distribution.

**Line 59:**

**… precipitation event.**

Modified as suggested.

**Line 74 (Comment):**

**I would rather more say "Scavenging of aerosol particles is not only affecting their number, but also their mass and other microphysical properties can change …**

 We have modified the sentence as Reviewer suggested.

**Line 98:**

**An Eulerian …**

This is now corrected.

**Line 99:**

**… arriving at the measurement site …**

 Modified as suggested.

**Line 101:**

**… advantage compared to Eulerian …**

Modified as suggested.

**Line 104:**

**…. Russia at the arctic …**

Modified as suggested.

**Line 110:**

**…. particle number size distribution …**

Modified as suggested.

**Line 123:**

**… environments …**

Modified as suggested.

**Line 124:**

**…. particle number size distribution observations …**

Modified as suggested.

**2          Data and Methods**

**2.1          Observations at SMEAR II, Hyytiala, Finland**

**Line 135:**

**Leave out "various".**

Word "various" is now removed as suggested.

**Line 135 (Comment):**

**Is this density realistic, seems very small, and does at least not represent any inorganic partitioning.**

The Reviewer is correct, the density of 1 g cm$^{-3}$ is relatively low for the particles observed in Hyytiälä. For example, in Häkkinen et al. (2012) particle density of 1.6 g cm$^{-3}$ was used based on the estimates given by Kannosto et al (2008). Kannosto et al. reported, for example, density values of 1.1-2 g cm$^{-3}$ for accumulation mode particles. Also, the density varies depending on the chemical composition of the particles (see e.g. Heikkinen et al., 2020) and thus any selected constant density over the whole measured size distribution would be an approximation even at its best. Initially, we wanted to use the unit density instead to achieve comparable

results with Tunved et al. (2013), as our methodology is partly identical. We also applied sensitivity analysis (see example figure below) with different densities, but same conclusions could be drawn.

We have now, however, used density of 1.6 instead which is closer to the real density of the particles in Hyytiälä. We have now modified the referred sentence followingly:

…spherical and had a constant density of $\rho$ = *1.6 g cm$^{-3}$ (see e.g. Häkkinen et al. 2012). Sensitivity analysis was conducted with unit density, 1 g cm$^{-3}$, following the approach in Tunved et al. (2013), but same conclusions could be drawn.*

[Figure]

***Example figure:*** *Total particle mass concentration derived from the DMPS measurements as a function of accumulated precipitation. Precipitation is binned into 0.5 mm bins, and for each the median concentration (black dots) and 25$^{th}$ and 75$^{th}$ percentiles (vertical bars) are then calculated. Subplot a) shows the concentration with density of 1 g cm$^{-3}$ and b) with density of 1.5 g cm$^{-3}$.*

**Line 145:**

**… was …**

This is now corrected.

**2.2          Trajectory calculations and airmass source analysis**

**-**

**2.3          Statistical mixed effects model**

**-**

**2.3.1          General description of the multivariate mixed effect model**

**-**

**2.3.2 Selection of relevant variables and determining the relative contribution of variable groups**

**-**

**3.   Results and Discussion**

**3.1          Effect of wet scavenging on the aerosol concentration**

**Line 273:**

**… number size distribution …**

 Modified.

**Line 279 (Comment):**

**Does Figure S6 now include particle mass of BC as stated in S7? This is now unclear to me, when discussing the ACSM data. Please clarify.**

We thank the Reviewer for noting this, as there is a mistake in the caption of Figure S7. The particle mass/sample size shown in Figs. S6 and S7 is based on the ACSM measurements only and does not include Aethalometer derived eBC mass. The caption of Fig. S7 is now corrected.

**Line 293 (Comment):**

**What do you mean by "when the air parcel it passing the cloud"? The cloud is built within the air parcel. Maybe, consider rewording this.**

We have now reworded this sentence (below). Please also see our reply to comment "**Line 34**".

*Lagrangian analysis demonstrates in-cloud scavenging occurring via the removal of activated aerosol particles from the atmosphere due to precipitation scavenging. However, this is the case only if the trajectory of the air parcel coincides with the conditions we define as "precipitating cloud".*

**Line 301 (Comment):**

**This is very hard to see from Figure 3. I suggest to place here a Figure that presents size fractions as (10 – 100nm, 100 – 300nm, etc.) as a function of accumulated precipitation in a two-dimensional plot.**

We thank the Reviewer for the comment, as also Reviewer#2 pointed out the same issue in the visualization. Based on the comments from Reviewer#2 we have added an additional plot (Fig. 3b) showing few selected particle number size distributions with 0, 2, 5, 15 and 20 mm of accumulated precipitation. The changes for the different sizes should me more clearly visible now. The corresponding figure (using GDAS derived trajectories) in the supplement was also updated.

 **General comment:**

**I find your argumentation in this section quite reasonable except from your statement in Line 305, where you say that the size of small activated particles decreases because of less competition with available supersaturation. Please clarify! Also, can you explain why number and mass concentrations are no longer decreasing at a certain set point of accumulated precipitation (see Figure 1)?**

When the concentration of accumulation mode particles is decreased in an air parcel because of activation and precipitation, the "remaining" accumulation mode in the consecutive cloud cycle along the travel of the air parcel (assuming relatively constant supersaturation conditions), produces less cloud droplets.

Hence, the remaining accumulation mode consumes less the available water vapour and smaller particles can activate. This is of course simplistic view assuming, for example, that the source of supersaturation remains constant. This, on the other hand, requires the meteorological conditions below the cloud base to remain constant over our trajectory transport region. In reality, the results of our analysis are the average of many experienced aerosol concentrations and cloud processing events. Anyhow, Figure 3 shows that largest changes

in size range relevant for activation, can be seen in very small precipitation values. Small precipitation values mean it is very likely that the precipitation takes place close to the measurements site (see example figure at the end of this reply). This indicates that the conditions are relatively close to the simplistic view. We have now clarified the text to make our statement clearer.

*We can qualitatively explain the observed behaviour in **Error! Reference source not found.Error! Reference source not found.** according to a simplistic view of the complex and highly dynamical process of activation. Assuming relatively constant meteorological conditions over our trajectory transport region, we can describe the precipitation cycle followingly: after the larger particles are removed through activation into cloud droplets, of which a subset will precipitate, the size of the smallest activated particles decreases in the consecutive cloud cycle, because of less competition for the available water vapour.*

A possible explanation for the observed asymptotic behaviour of the particle mass/number concentrations after certain amount of accumulated precipitation (Figure 1) could be the local emissions, or emissions relatively close to the observation site. As we are inspecting the particle mass/number in Lagrangian framework, we see how the particles are affected along the airmass route. However, local emissions could dominate the concentrations at high accumulated precipitation values, and thus no further decrease is seen in particle mass/number even with increasing accumulated precipitation (local sources produce significant amounts of particles even though arrived airmasses had experienced lot of precipitation during travel). Similar behaviour, particle mass achieving asymptotic behaviour with higher values of accumulated precipitation, has been observed for Arctic location (Tunved et al., 2013) and in tropics (Dadashazar et al., 2021). These are mentioned in the introduction. We now also shortly state this when discussing the results in Section 3.1.

*Particle mass decrease reaches asymptote after ~10 mm of accumulated precipitation. This could be due to local sources producing significant amounts of particles even though arrived airmasses have experienced large amounts of precipitation during travel. Similar behaviour has been observed for Arctic location and in tropics (Tunved et al., 2013; Dadashazar et al., 2021).*

[Figure]

*Example figure: Subplot (a) shows total particle mass concentration (assumed unit density) derived from the DMPS measurements as a function of accumulated precipitation. Precipitation is binned into 0.5 mm bins, and for each the median concentration (black dots) and 25th and 75th percentiles (vertical bars) are then calculated. Subplot (b) shows the corresponding number of precipitation events as a function of accumulated precipitation and subplot (c) the average distance from the SMEAR II station.*

**3.2        Effect of wet scavenging on the aerosol composition**

**Line 347:**

**… components …**

This is now corrected.

**General comment:**

**The scattering in Figure 4d might be largely due to the fact of local contribution of aerosols during the warmer months. The analysis idea that you apply and which is in general very nice does then fail as the accumulated precipitation is not connected to the mass concentrations you measure here as fresh and local emissions are dominating. For the cold months I agree with your explanation that the aerosol is well internally mixed because of the long-range transport character leading to similar removal rates for the different species.**

The referee is correct here, strong local contribution can distort our analysis. We have now modified the sentences were this is discussed also based on the comments from Reviewer#2 to make this clearer:

*$SO_4$ seems to be removed less efficiently… This is surprising as sulfate is more hygroscopic than Org and eBC. There are two possible explanations for the observed differences. First, this could indicate …. Second, the stronger contribution of local sources of $SO_4$ during warm months could distort our analysis and result in lower derived removal efficiency. This could be caused either by local $SO_4$ sources being stronger in summer compared to winter or the long-range airmasses are arriving from locations with significantly less SO4 or its precursors.*

**3.3 Effect of in-cloud processing on aerosol concentrations and composition**

**Line 531 (Comment):**

**Check bracket types.**

Our aim here was to use the mathematical notation for the range/interval we show. Thus, brackets of type "[]" indicate closed range, i.e. [3, 30] means both 3 and 30 nm particles are included in the range/interval (closed interval). Brackets of type "()" indicate open interval, so for example the notation (30, 50] indicates the particles with diameter exactly of 30 nm are not included in the range whereas particles of 50nm are included (half-open interval). This is very common way for writing/noting intervals in mathematics textbooks, and this notation type is following the international standard ISO 31-11 for mathematical signs and symbols for use in physical sciences and technology. Please see this wiki-page as an example: https://en.wikipedia.org/wiki/ISO_31-11

**Line 531:**

**… clearest in size ...**

Corrected as suggested.

**Line 563:**

**… following …**

The word "followingly" is now completely removed from the figure captions and replaced with ":" instead.

**General comment:**

**Figure 9 needs in my view more clarification. For the smaller size fractions there seems to be also a mass decrease from group nr.1 to group nr. 3. Can you comment on this?**

This is likely due to the cloud processing: smaller particles take up water in supersaturation conditions and this in combination with aqueous phase reactions can lead to the size increase. In addition, smaller un-activated

particles can collide with cloud droplets leading to the concentration decrease of un-activated particles (e.g. Romakkaniemi et al. 2006).

The scope of our study was to investigate accumulation mode sized particles, which are the most relevant for cloud processing. Thus, it is not possible with our current methodology to provide conclusions or detailed reasoning for the smaller sizes. Investigation of how precipitation (and other related processes) affect particle number and mass concentrations for particles of smaller sizes would be great topic for future research, but it is unfortunately out of the scope of our study. Thus, we do not dare to speculate more on the processes related to particles of smaller size.

**4 Conclusions**

**Line 568:**

**… affect the …**

The sentence is now modified.

**Line 582:**

**… effect on …**

Modified as suggested.

**Line 597:**

**… airmass trajectory …**

Modified to …conditions along the airmass *trajectories* affect the observed…

**REVIEWER#2**
**General comment**

**The study by Isokääntä et al., investigates the effect of cloud processing and wet removal on aerosols, using airmass history analysis and in situ measurements from a boreal forest in northern Europe. The manuscript is interesting and the quality of the analysis is good but the interpretation of their results not always convincing. Additionally, the structure of the manuscript should be improved for clarity. I think this manuscript is within the scope of ACP but the following points should be addressed before I can recommend its publication.**

We thank the Reviewer from the helpful comments and suggestions which have greatly improved the clarity and message of our manuscript. Please see our point-by-point answers below.

**Major comments**

**Multivariate mixed effect model**

**A good part of section 2 is dedicated to describing this statistical model but then its results are not really used in the rest paper. There are only a few short parts referring to the model in the main result section. It should be clear what is the purpose of the model and how it contributes to the findings of this paper. I think the paper would be clearer and more concise without the model (which could be used for a separate publication). It is fine for me if the authors decide to keep the model in the paper, but then they should describe its results in a clear way and discuss them coherently with the other findings.**

It is true that the results from the statistical modelling are not fully exploited in our study. Unfortunately, for example, detailed interpretation of all the regression coefficients would be out of scope of our current focus. Instead our aim was to use the model results to support our conclusions.

We have now significantly reduced the text describing the model in section 2.3 to improve the text flow and comply with the current interpretations of the model results in the manuscript. Part of the technical details that are necessary for the understanding the model, are now given in Appendix A. Text describing the variable selection is moved to the to the supplementary material. In addition, the subsections 2.3.1-2.3.2 are now removed as they were not necessary after the modifications.

We wanted to keep the model results in this manuscript as they are supporting our results derived by other methods. Even though we have now also added statistical tests to confirm the differences we see are statistically significant, it is still beneficial to observe that our results persist when more of the factors affecting particle mass are considered.

**Effect of wet scavenging on aerosol concentration**

**In line 284 the authors say that the vertical position of the air parcel with respect to the cloud is not considered. So, what if the air parcel is actually above the precipitating cloud? This is a potential drawback which should be addressed to make the analysis more robust and convincing. ERA-interim contains vertically resolved information concerning the cloud fraction and type, so it should be possible to distinguish between an airmass below, inside or above the clouds. I urge the authors to include this type of analysis in their work. With this addition, it should also be possible to compare more directly in-cloud with below-cloud scavenging, and confirm if the first is more efficient than the latter as currently hypothesized in the manuscript.**

Yes indeed, the "2D precipitation" (precipitation at the surface) data is a drawback in our study. However, the RH data we use as a cloud proxy is 3D i.e. it is defined at the height of the air parcel. This was mentioned in the methods section in lines 193-195 (original manuscript).

Due to the precipitation data being 2D (see e.g. HYSPLIT user guide: https://www.arl.noaa.gov/documents/reports/hysplit_user_guide.pdf or Berrisford et al., 2011), it is still not possible to define the vertical position of the air parcel relative to precipitating cloud (i.e. if the air parcel is in the precipitating cloud or if the precipitating cloud is above or below of our air parcel). The limitations associated with applying 2D precipitation data to this type of analysis is something that we have made an effort to highlight in this study, as this was not always been made clear in earlier studies (Tunved et al., 2013; Kesti et al., 2020; Dadashazar et al., 2021). At the moment we are working on obtaining 3D precipitation fields from large scale models, but this is currently ongoing and out of the scope of this manuscript.

Using cloud fraction data from ERA-Interim could provide us information whether our trajectory is either above or below the cloud (as currently we know if it is "within the cloud") but it does not remove the problem with precipitation. Even if we would have vertically resolved cloud data, we are still limited with the 2D precipitation data. For example, the problem that precipitation from the cloud could evaporate before reaching the ground persists. Multiple cloud layers could also exist, but we are not able to resolve which one of those is the precipitating one.

We have noted that the wording we used on line 285 "trajectory position respect to cloud" might be confusing, as we are discussing specifically precipitation scavenging. We have now reworded the text: "*trajectory position with respect to precipitating cloud*", as we do have information of the vertical position of clouds in general due to our RH proxy for clouds being at the height of the airmass. In addition, reference to methods section is added to this sentence.

We have added few more sentences to the methods section to highlight the 2D nature of the precipitation data even more.

*In addition, since the precipitation data is not vertically resolved, it is possible that the air parcel is above the precipitating cloud, and thus not affected by the precipitation. Other possible scenario would be a case where our airmass is below the precipitating cloud, but precipitation evaporated before influencing our air parcel. This is an unfortunate limitation in this type or analysis and may contribute to the variability of the results. Despite, successful analyses have been conducted recently (Dadashazar et al., 2021; Kesti et al., 2020).*

**Effect of wet scavenging on the aerosol composition**

**Line 357: "SO4 is removed less efficiently than Org and eBC", this is counterintuitive since sulphate particles are more hygroscopic compared to organics and black carbon. Hence, they should activate and be removed preferentially in clouds. The authors draw this observation from the results in Figure 4d, which shows the normalized mass concentration as a function of accumulated precipitation. However, the data in the figure look odd because the organics and eBC normalized mass is less than 1 for an accumulated precipitation value of zero. Can the authors explain this discrepancy? Just by looking at the relative decrease it seems like sulphate is scavenged as efficiently as organics and eBC, so the authors should maybe revise their statement.**

We thank Reviewer for noting this, as we realized that the Figure4 can be misinterpreted. At zero accumulated precipitation, the points for organics, eBC and SO4 are on top of each other's and thus eBC and organics are not visible. We have now modified the figure (4d and 4e) by adjusting the size and shape of the markers to make them all visible also when accumulated precipitation is zero.

In addition, our unprecise wording caused the confusion related to the interpretation of figure 4d/e and the precipitation removal efficiencies of different constituent. Reviewer is correct for saying sulfate particles being more hygroscopic compared to organics and black carbon. Hence sulfate should be scavenged more efficiently than e.g., BC. There are two possible explanations for the unexpected observed differences. First, this could indicate that more of the SO4, compared to Org and eBC, is distributed to smaller particles during warmer months which reduces both CCN activation potential and thus removal of activated particles by

rainfall. Second, stronger contribution of local sources of SO4 during warm months could dominate the SO4 concentrations in Hyytiälä distorting our analysis and resulting in the observed behaviour. This could be caused either by local SO4 sources being stronger in summer compared to winter or the long-range airmasses are arriving from locations with significantly less SO4 or its precursors.

We have now modified our sentences to make our interpretation clearer:

*SO4 seems to be removed less efficiently… This is surprising as sulfate is more hygroscopic than Org and eBC. There are two possible explanations for the observed differences. First, this could indicate that more of the $SO_4$, compared to Org and eBC, is distributed to smaller particles during warmer months which reduces both CCN activation potential and thus removal of activated particles by rainfall. Second, the stronger contribution of local sources of $SO_4$ during warm months could distort our analysis and result in lower derived removal efficiency.*

**Effect of in-cloud processing on aerosol concentrations and composition**

**The choice of a threshold RH equal to 94% is arbitrary, the authors should do some type of sensitivity study to support their choice. For example, they could explore how the results change when a smaller and larger threshold are used (e.g. 90% and 98%).**

We did apply different thresholds during the data analysis to see if our results are sensitive to the selected value. The conclusions of our study are the same with RH limits of 85, 90, 94 and 98 (e.g. larger SO4 masses are observed for airmasses with experience of non-precipitating clouds compared to airmasses with history of clear skies). Most significant difference is related to number of observations for the strictest defined wet processing group (group 3, see e.g. Fig. 5). With lower limit (85, 90) the number of observations in this group increases, whereas with higher limit (98) it decreases.

The final limit of 94% was selected as it has been reportedly used in earlier study with similar approach (Tunved et al., 2004). It is also very close to the critical RH limits used, for example, in both reanalysis data and models for cloud formation. For ERA-Interim, for example, the cloud fractions are partly derived from RH values between 80 % to 100 % with values increasing towards surface (Tiedtke, 1993; Dee et al., 2011). Additionally, in MPI-ESM (ECHAM6.3) the limit is dependent on the height and has been given values between 90%-96.8% close to the surface (Mauritsen et al., 2019). In CSIRO climate model the RH limit for cloud formation between two lowest levels is 95% (Gordon et al., technical report).

We have now moved the paragraphs explaining our selection from the beginning of section 3.3 into the methods section 2.2 and provided more justification for our selection of the RH value.

*The relative humidity at the altitude of the airmasses was used as a proxy for in-cloud cases. We selected a limit of RH > 94 % (as in Tunved et al., 2004) for which we assume the airmass is inside a cloud or fog (we do not separate these cases). We would like to note that even if the approximation for the in-cloud cases is not very accurate based on the RH values only, the humidity in these cases is high enough for the particles to have taken up significant amounts of water. Strong hygroscopic growth can be observed before activation, and e.g. for inorganic salts the deliquescence RH is well below 94 % (e.g. Cruz and Pandis, 2000; Zieger et al., 2017; Lei et al., 2018). Thus, it is safe to assume the aqueous phase processes, whether in cloud or inside fogs, are taking place when RH of 94 % is exceeded. The selected limit for the in-cloud cases is relatively close to the values used for critical RH for cloud formation in reanalysis data and large-scale models. For example, in ERA-Interim values between 80 % to 100 % with increasing values towards the surface are used (Tiedtke, 1993; Dee et al., 2011). In MPI-ESM model (ECHAM6.3), the limit has been given values between 90 % to 96.8 % close to the surface (Mauritsen et al., 2019). Sensitivity analysis was conducted with RH limits of 85 %, 90 % and 98 %, but same conclusions could be drawn.*

**Moreover, while reading the manuscript for the first time, I was confused about why the authors decided to give the same weight to every air mass within cluster 3 and did not consider the time spent inside the cloud (I had to read through the entire section before finding an answer).**

We assume that by cluster 3 the Reviewer now means the wet processing group 3 including observations having experienced only non-precipitating clouds during the last 24 hours. It is true that we do not separate/weight by number of hours in the non-precipitating cloud. Our definition of "only non-precipitating clouds" (Table 1 in the revised MS for trajectory availability) is strict by nature, as we wanted to exclude the effects of precipitation from this in-cloud analysis. Unfortunately further separation of this group would not be possible, as when the trajectories are collocated with the measurements, we are further limited by the availability of these observations. Further division of group 3 could maybe be possible with the DMPS measurements, as they exist for longer time period, but we wanted to be consistent and use the same groupings for size distribution and chemistry measurements.

Please see our response to comment "**Lines 414-416**". We have now added a sentence to clarify this (near to the beginning of section 3.3, where the grouping shown in Table 1 is used).

Please also see our reply for the comment below, where we explain that our aim was to first look at the effect of non-precipitating clouds specifically (bar plots, now boxplot-like, starting with Fig 5) and then later we shortly discuss how the time the airmass has spent inside clouds (all clouds) affects the mass fractions (Fig 8).

**I think that Figure 8 highlights the effect of cloud processing much better than Figure 5 and 6 because it clearly shows that the sulphate mass increases with time spent at RH>94%. Hence, I would recommend to show the same also for total particulate mass and number concentration and start the discussion from there. If this is done then figure 5 and 6 could probably be removed but I leave the final decision to the authors.**

We wanted to inspect the in-cloud processing with two approaches. First, we wanted to specifically inspect only those clouds along the trajectories which did not show simultaneous precipitation (Figures 5 and 6). Due to this (only clouds, no precipitation) being a relatively strict limitation (Table 1 in the updated manuscript), we only looked 24 hour long trajectories to have enough overlapping cases with the composition measurements (with increasing the trajectory length, the number of non-precipitating trajectories decreases).

For Figure 8 we did not apply any distinction relative to precipitation and thus the time spent in RH>94% (=time "in-cloud") includes all cases from the whole 96 hour long trajectory. This is now explicitly stated in the manuscript. Due to the limitations in the composition measurements, this distinction (using only non-precipitating clouds) can not be made, as we would lose a lot of cases with longer exposure times for in-cloud conditions. Therefore we included all clouds (precipitating and non-precipitating) for this analysis. The drawback of presenting data in this way is the fact that two processes (precipitation scavenging and in-cloud processing) are mixed: when inspecting the change in the absolute SO4 mass (additional figure in the updated SI material, Fig. S13), we can see that the increase in the mass is seen only with long exposure times in RH>94%. This is due to the fact that the trajectories can also have experienced precipitation, which, in contrast, decreases the SO4 mass.

*When investigating the composition of the particles as a function of time in RH > 94 % (**Error! Reference source not found.**) when no distinction relative to precipitation is applied (i.e. time in cloud can also include precipitating clouds), we observe an increase in sulfate mass fraction with time spent under the high humidity conditions.*

**The classification used in this section (WC, WP, CC and CP) is more specific compared to the previous sections, where only warm and cold periods were separated. It is not clear why the authors decided to use two different classification and I would recommend to use the same classification throughout the manuscript.**

We wanted to start simpler and then add the extra dimension later when the sulfate formation in clouds is discussed. The clean/polluted division was to see differences between the airmass origins i.e. how initial SO2 affects to the SO4 formation. However, all 4 groups cannot be used in the earlier section (3.2) as we don't have enough observations to further divide the composition data when inspecting the precipitation along the whole trajectory.

The reason why we wanted to add the clean/polluted division (in addition to our two seasons, cold and warm) was to confirm that we are not just looking at an "airmass source" effect i.e. high RH would (randomly) occur for those airmasses that come from highly polluted areas. This could, in theory, be possible due to meteorological reasons (i.e. more cloud cover for trajectories arriving from specific region/direction).

Please also see our reply to comment for "**Lines 473-477**".

We added a sentence to the manuscript (original draft lines 434 ->) to clarify this:

… source areas to different meteorological conditions (i.e. group 1 vs group 3). *This type of source area artefact could take place, for example, if cloud occurrence would be more frequent for airmasses arriving from certain directions, which could (randomly) coincide with higher SO$_4$ observations*. Further justification for our choice of these sectors…

Please also note that we had a typo in our plotting script for Figure 9 in the original manuscript, and thus b-panel (sector WC) was incorrectly displayed. This is now corrected.

**When comparing changes between groups it would be important to check if these differences are statistically significant, I would encourage the authors to run a simple statistical test to support their observations (I suggest to use a nonparametric test like the Mann-Whitney U rank test)**

We thank the Reviewer for noting this. We have now applied the Mann-Whitney U rank test to explore if the differences we observe are statistically significant. The results (p-values) are reported in tables in the supplementary material (Tables S4, S5 and S6) and mentioned in the text when discussing the results.

A reference (below) is also added for the used test.

H. B. Mann. D. R. Whitney. "On a Test of Whether one of Two Random Variables is Stochastically Larger than the Other." Ann. Math. Statist. 18 (1) 50 - 60, March, 1947. https://doi.org/10.1214/aoms/1177730491

**Minor comments**

**Line 62: I would mention that not all particles activating as cloud droplets actually precipitate..**

This is now mentioned in the text:

*The in-cloud scavenging efficiency is controlled by nucleation (i.e. aerosol activation) and impaction scavenging. It is dominated by activation of aerosol particles into cloud droplets (e.g. Ohata et al., 2016), from which a fraction precipitate.*

**Line 81: I would include also the study of Lamkaddam et al., 2021 as a reference for aqSOA formation, it is an important experimental study.**

We thank the Reviewer for noting this. The reference is now added.

*The production of secondary organic aerosol through aqueous phase processes (aqSOA) has been also reported (e.g. Ervens et al., 2011; El-Sayed et al., 2015; Ervens et al., 2018; Mandariya et al., 2019;*

*Lamkaddam et al., 2021). It has been suggested that aqSOA formation is comparable in magnitude with SOA formation through gas phase oxidation processes (Ervens et al., 2011).*

**Line 122: this study is focusing on more than just "the influence of below-cloud scavenging during transport", so this sentence should be reformulated.**

The referred sentence is now reformulated (below).

To explore the influence of below-cloud scavenging *and aqueous phase processing in-cloud* during transport on observed aerosol size distribution and chemical composition in biogenically dominated environments…

**Line 280-281: the particle number concentration tends to increase for some bins with an accumulated precipitation above about 30mm, do the authors have an explanation for this effect? Is it just statistical noise or a real signal? It would be interesting to see if the effect persists also when using larger bins.**

The effect of slight increase in the particle number concentration with high values of accumulated precipitation also seems to persist when using larger bins (see figure below with bins of 1 mm). We assume this effect can mainly be explained by statistical noise as we have much fewer observations for higher accumulated precipitation. In addition, there is lot of variation in the median values (black dots) with higher values of accumulated precipitation, compared to the ones prior accum.precip = 10-20 mm.

[Figure]

*Example figure: Total particle (dp = 3-1000 nm) number (a) concentration and (b) number of observations per bin as a function of 0-50 mm accumulated precipitation along the 96-hour HYSPLIT airmass trajectories. The black dots in (a) show the median values and bars highlight the 25th-75th percentiles for each 1 mm bin of accumulated precipitation. The figure includes DMPS data between January 2005 and August 2019.*

**Lines 311-314: It is difficult to assess if sub100nm particles are really unaffected by precipitation as stated here because of the way the data are shown. I would suggest to show the decrease of particle number in different size bins as a function of accumulated precipitation (something similar to Figure 1b, maybe using larger bins). This additional figure could be included in the supplement.**

Figures showing the changes in the smaller size ranges with 1 mm precipitation bins is shown below (the number concentrations are now added into the supplement). We can immediately see that for smaller size ranges (a-b), we don't see clear decrease in the number/mass concentration, but with increasing the size (c), the initial decrease becomes more prominent. For the smaller sizes, we can see small initial increase which could be e.g. due to precipitation enhancing the NPF event frequency with some delay, but this investigation is out of scope for our study.

We have now reworded our sentence slightly to avoid stating the small sizes are completely unaffected.

*As shown in **Error! Reference source not found.Error! Reference source not found.**, the aerosol concentrations in size range of 10-50 nm show no clear sensitivity (decrease) to accumulated precipitation and the largest decrease in concentrations are shown in size range of $d_p$ ~ 100 nm and above suggesting that in-cloud scavenging is the dominating removal mechanism in the submicron particle size range in the studied environment. Inspection of selected size ranges (Figure S8) confirms that only larger sizes start exhibiting the decrease as a function of the accumulated precipitation.*

[Figure]

***Example figure:*** *Number and mass concentrations with 1 mm bins for accumulated precipitation for different size ranges.*

**Figure 3: for better visualization I would also extract a few representative size distributions, with the colormap is not always easy to appreciate changes (showing for example the average PSD corresponding to 0mm, 20mm and 40mm accumulated precipitation)**

We agree with the referee that showing few representative size distributions helps in the visualization. These are now added to the plot (shown also below).

[Figure]

**Lines 359-361: what is the meaning of "This could indicate that more of the SO4, compared to Org and eBC, is distributed to smaller particles during warmer months which reduces both CCN activation"? Sulphate is more hygroscopic than organics and eBC, so if smaller particles contains a larger fraction of sulphate then they are more likely to activate as CCN. In general, this study does not provide any evidence that smaller particles contain a larger fraction of sulphate, so I would consider removing this sentence or to back it up with previous studies if available.**

We agree with the Reviewer that our sentence related to the distribution of sulfate to the smaller particle sizes is speculative. In the manuscript we give two speculative explanations to the observations shown in Figure 4d as explained in an earlier reply. Unfortunately we don't have data allowing to draw conclusion to the observed and, as pointed out by the Reviewer in the earlier comment, unexpected behaviour shown in Figure 4d. Hence, we feel that the speculative explanation of the observation should be given. We have modified the text to make the speculative nature of our sentence clearer.

Please see our response to the major comment on "**Effect of wet scavenging on the aerosol composition**" and the modified manuscript section therein.

**Lines 373-374: For me the most striking evidence that wet removal in winter is more efficient compared to summer is the comparison of figure 4d and 4e, so I would start from there. The results of the statistical model are less obvious and I would mention them afterwards (if the authors decide to keep the model in the manuscript).**

We have now moved the referred sentence further down in the text as the Reviewer suggested. We decided to keep the model (please see our response to the first major comment) but have moved the details of the model into the appendix and supplementary material.

**Figure 4: are panel a,b and c relative to both cold and warm periods? I would mention this in the caption for clarity.**

Yes, panels a, b and c have data from the whole year. We have now modified the figure caption to include this information.

**Line 408: what is the meaning of "figs" here?**

We thank the Reviewer for noting this, as there is a typo in the text. The word should be "fogs" instead. This is now corrected.

**Lines 414-416: It took me a while to understand why restricting the time window to 24 hours would increase the amount of observations per group. Consider rephrasing or explain this better.**

We have added more clarification to make this clearer for the reader.

*With longer trajectories, more of the trajectories would contain precipitating clouds, which would lead to a reduction of observations in group 3. Sensitivity analysis was conducted by limiting airmass experience to 36 and 48 hours, but same conclusions were achieved.*

**Line 460: include 1 or 2 sentences explaining what were the main findings of Riuttanen et al., 2013, otherwise it is difficult to follow the comparison here.**

We thank the Reviewer from the suggestion. Some of the main findings from Riuttanen et al are now described in the text.

*Riuttanen et al (2013), conducted trajectory analysis to investigate trace gases observed in SMEAR II, and our temperature-based division coincides well with the seasonality of $SO_2$ concentration. They concluded, for example, that combustion related $SO_2$ is mainly transported to SMEAR II from Eastern Europe during winter months. In addition, for high particle concentrations arriving to SMEAR II, they observed the airmass origins to be dependent on particle size.*

**Lines 473-477: the analysis described here is not very clear for me. The authors are looking at the effect of aqueous phase processing on sulphate production and it should not matter if the sulphate is coming from anthropogenic or biogenic sources, so why making this distinction? Additionally, Figure S17 does not prove anything concerning the effect of marine emissions: it simply shows that cluster 1 is typically made of air masses from the boundary layer whereas air masses in clusters 2 and 3 are more often coming from the free troposphere. This is still an interesting observation but it should just be presented for what it is. Finally, it is important to note that sulphate from DMS oxidation is also mainly added via aqueous phase processing (e.g. Chen et al., 2018; Hoffmann et al., 2016), so there is no simple way to distinguish between sulphate from marine and anthropogenic emissions simply based on the air mass altitude as done here.**

By the analysis we wanted to show that the increase in sulfate concentrations is driven by sulfate formed mostly in the clouds and not just directly above the sea surface derived from DMS emissions. Our intention was not to distinguish sulfate from marine emissions and anthropogenic emissions. We have now clarified this in the text.

*Further, to show that the increase in sulfate concentrations is driven by sulfate formed mostly in the clouds and not just directly above the sea surface derived for example, from dimethyl sulfide emissions (see e.g. Barnes et al., 2006), we investigated the vertical transport of the airmasses. This analysis showed no evidence (see Section S4) that this type of transport is significantly influencing the results presented here.*

**Figure 5 and following: why did you decide to show the confidence interval from bootstrap replicates instead of the interquartile range as in Figure 4? I tend to prefer the IQR to provide an idea of the underlying variability.**

Initially we wanted to use bar plots to show our data as they show the differences between medians more clearly. However, bar plots are not practical when wanting to show IQR, and thus we used confidence intervals instead. However, as the referee points out, we do use IQR in the other plots. For consistency, we have now changed all bar plots (Figures 5, 6 and 9 in the original manuscript and Figures S11, S14, S23, S24 and S27 in the original supplementary material) into plots showing the medians as horizontal lines and boxes extending

to Q1 and Q3 (IQR). To show the differences in medians more clearly in this approach, we have now also included the medians as numerical values in these plots. Figure captions have also been modified accordingly.

**Lines 504-505: There is probably a mistake here: the oxidation of organics decreases their volatility (e.g. Kroll & Seinfeld, 2008).**

Reviewer is correct, oxidation can also decrease the volatility of organic molecules. However, fragmentation will lead to a decrease in molecular mass and thus the volatility can increase (Kroll et al., 2011, Figure 1), and this was the situation we wanted to refer to in the text (i.e. organics become oxidized to volatile compounds within cloud droplets, see Ervens et al., 2018). We have now modified the referred sentence to avoid any misinterpretations.

*It has also been suggested that water soluble SOA (originating from other sources than aqueous phase processing) in the cloud droplets can become oxidized to form more volatile compounds leading to evaporation. This could lead to a decrease in total SOA mass, even though additional aqueous phase SOA mass is formed (Ervens et al., 2018).*

**Lines 507-514: this paragraph is confused because two different interpretations are mixed together. In particular, the authors first say that aqueous phase production of SOA is negligible and then that the increased mass, produced via aqueous phase, is compensated by reduced production in the gas phase due to lower solar radiation levels. These two things are different and the authors should clarify what is their interpretation of the results. It should be possible to look at the effect of solar radiation by comparing air masses with a similar level of insulation.**

We well understand the confusion of the reviewer. The point we try to make here is that when the total organic mass is considered (in the studied environment), the local photochemically driven SOA production dominates over the aqueous phase SOA formation. Because of this, we don't see aqueous phase chemistry driven increase in total org mass when groups 1 and 3 are compared. We have now clarified this in the text.

*Our results indicate, that in the boreal forest dominated Northern Europe the photochemical SOA formation in the gas phase dominates over SOA formation in the aqueous phase, when the total organic mass is considered. This applies for both warmer and colder seasons, and in the case of clean and polluted airmasses.*

**Lines 539-540: Could the authors try to calculate a simple mass balance and see if the increase in these size fractions matches with the sulphate increase? This simple calculation could help supporting the conclusions reported here.**

We thank Reviewer for suggesting this. We have now conducted simple calculations to see how the mass increase observed for SO4 from ACSM (Figure 6 in the updated MS) compares to that derived from DMPS data for particles with diameters larger than 200 nm (Figures 9 and S16 in the updated MS/SI). For this we assumed the increase, in DMPS size classes > 200 nm, is only due to increase in sulfate mass. The mass increases derived from the DMPS data are 2-3 times larger depending on the sector (WC, WP, CC, CP) when compared to the increase in SO4 mass derived from the ACSM.

However, there are some issues related to this type of calculation in our study. The particles, that are growing due aqueous phase reactions, are not only consisting of SO4. We can, for example, have fully organic particle, into which the SO2 then condenses forming SO4 and thus increases the size of the particle and moves these particles from smaller size bins to larger size bins. Unfortunately, the DMPS is not capable of differentiating this (chemical composition). Thus, the mass increase we see in the DMPS data, also includes, in this example, also the "original" organic mass plus the added mass from SO4. Therefore, the mass increase observed for DMPS data can be larger than the increase seen solely for SO4 based on the ACSM data.

In addition, the instruments differ also by technical means. For example, ACSM measures the particle mass, but in DMPS, the number concentration is measured, and mass achieved by assuming constant density over the whole size range.

Therefore, the changes in mass concentrations from DMPS are not directly comparable with the changes we see in the chemical composition. However, we believe they can still be used indicatively, as we have done. This should be clear from the sentence we have in the manuscript (not modified) in the end of Section 3.3:

*These results suggest that the SO₄ formed via in-cloud aqueous phase processes is mainly distributed to particles having a dry diameter between 200 - 1000 nm.*

**Figure 7: I would recommend enlarging the map (reducing the geographical region) for a better visualization.**

We have now modified the map to show reduced geographical region as the Reviewer recommended.

**Figure 8: I would include in this figure also the absolute concentration trends.**

We have now added the figure in the supplementary material (also below) and added a sentence referring to it into the manuscript:

*Inspection of the absolute mass of the species (Figure S14) also shows an increase in SO4 mass with longer exposure times in RH>94 %, whereas decreases in mass of all species is seen with increasing time of experienced precipitation.*

[Figure]

***Example figure:*** *The mass concentrations of Org, SO₄ and eBC for clean and more polluted airmasses as a function of time spent in RH > 94 % (a and c) and in precipitation (b and d). Figure shows median values for each 1-hour bin, if 10 or more data points were available in the bin. The figure is based on observations between March 2012 and August 2019.*

**Lines 600-603: I would simply say that an increase in the organic mass due to aqueous phase processing was not observed (this study did not differentiate between different types of organics).**

We have now replaced the referred sentences with "*Increases in the total organic mass due to aqueous phase processing was not observed.*" as the Reviewer suggested. The language in the whole conclusion section has also been improved for clarity.

**Table S1: I would mention in the table if the data are publicly available and provide a direct link if possible.**

The data are publicly available as stated in the Data availability section. The SmartSMEAR database (link provided in the Data availability section) provides most of the variables but unfortunately the user must manually select the station and variable for downloading and thus no direct link to each variable can be provided. The chemical composition data (ACSM and Aethalometer) is also publicly available through EBAS (link provided in the Data availability section), but also there the user must select location and variable to be able to download the data.

**References**

Berrisford, P, Dee, DP, Poli, P, Brugge, R, Fielding, M, Fuentes, M, Kållberg, PW, Kobayashi, S, Uppala, S, Simmons, A.: The ERA-Interim archive Version 2.0, ERA Report series, Shinfield Park, Reading, https://www.ecmwf.int/node/8174

Dadashazar, H., Alipanah, M., Hilario, M. R. A., Crosbie, E., Kirschler, S., Liu, H., Moore, R. H., Peters, A. J., Scarino, A. J., Shook, M., Thornhill, K. L., Voigt, C., Wang, H., Winstead, E., Zhang, B., Ziemba, L., and Sorooshian, A.: Aerosol responses to precipitation along North American air trajectories arriving at Bermuda, Atmos. Chem. Phys., 21, 16121-16141, 10.5194/acp-21-16121-2021, 2021

Dee, D. P., Uppala, S. M., Simmons, A. J., Berrisford, P., Poli, P., Kobayashi, S., Andrae, U., Balmaseda, M. A., Balsamo, G., Bauer, P., Bechtold, P., Beljaars, A. C. M., van de Berg, L., Bidlot, J., Bormann, N., Delsol, C., Dragani, R., Fuentes, M., Geer, A. J., Haimberger, L., Healy, S. B., Hersbach, H., Hólm, E. V., Isaksen, L., Kållberg, P., Köhler, M., Matricardi, M., McNally, A. P., Monge-Sanz, B. M., Morcrette, J.-J., Park, B.-K., Peubey, C., de Rosnay, P., Tavolato, C., Thépaut, J.-N., and Vitart, F.: The ERA-Interim reanalysis: configuration and performance of the data assimilation system, Quarterly Journal of the Royal Meteorological Society, 137, 553-597, https://doi.org/10.1002/qj.828, 2011.

Chen, Q., Sherwen, T., Evans, M., & Alexander, B. (2018). DMS oxidation and sulfur aerosol formation in the marine troposphere: a focus on reactive halogen and multiphase chemistry. Atmospheric Chemistry and Physics, 18(18), 13617–13637. https://doi.org/10.5194/acp-18-13617-2018

Ervens, B., Sorooshian, A., Aldhaif, A. M., Shingler, T., Crosbie, E., Ziemba, L., Campuzano-Jost, P., Jimenez, J. L., and Wisthaler, A.: Is there an aerosol signature of chemical cloud processing?, Atmos. Chem. Phys., 18, 16099–16119, https://doi.org/10.5194/acp-18-16099-2018, 2018.

H.B. Gordon, L.D. Rotstayn, J.L. McGregor, M.R. Dix, E.A. Kowalczyk, S.P. O'Farrell, L.J. Waterman, A.C. Hirst, S.G. Wilson, M.A. Collier, I.G. Watterson, and T.I. Elliott. The CSIRO Mk3 Climate System Model. CSIRO Atmospheric Research Technical Paper No. 60. https://publications.csiro.au/rpr/download?pid=procite:ff94db7e-ad41-40bf-b6be-2ab1ad07805c&dsid=DS1

Heikkinen, L., Äijälä, M., Riva, M., Luoma, K., Dällenbach, K., Aalto, J., Aalto, P., Aliaga, D., Aurela, M., Keskinen, H., Makkonen, U., Rantala, P., Kulmala, M., Petäjä, T., Worsnop, D., and Ehn, M.: Long-term sub-micrometer aerosol chemical composition in the boreal forest: inter- and intra-annual variability, Atmos. Chem. Phys., 20, 3151–3180, https://doi.org/10.5194/acp-20-3151-2020, 2020.

Hoffmann, E. H., Tilgner, A., Schrödner, R., Bräuer, P., Wolke, R., Herrmann, H., Hans, E., Tilgner, A., Schrödner, R., Bräuer, P., Wolke, R., Herrmann, H., Hoffmann, E. H., Tilgner, A., Schrödner, R., Bräuer, P., Wolke, R., & Herrmann, H. (2016). An advanced modeling study on the impacts and atmospheric implications of multiphase dimethyl sulfide chemistry.Proceedings of the National Academy of Sciences of the United States of America, 113(42), 11776–11781. https://doi.org/10.1073/pnas.1606320113

Häkkinen, S. A. K., Äijälä, M., Lehtipalo, K., Junninen, H., Backman, J., Virkkula, A., Nieminen, T., Vestenius, M., Hakola, H., Ehn, M., Worsnop, D. R., Kulmala, M., Petäjä, T., and Riipinen, I.: Long-term volatility measurements of submicron atmospheric aerosol in Hyytiälä, Finland, Atmos. Chem. Phys., 12, 10771–10786, https://doi.org/10.5194/acp-12-10771-2012, 2012.

Kannosto, J., Virtanen, A., Lemmetty, M., Mäkelä, J. M., Keskinen, J., Junninen, H., Hussein, T., Aalto, P., and Kulmala, M.: Mode resolved density of atmospheric aerosol particles, Atmos. Chem. Phys., 8, 5327–5337, https://doi.org/10.5194/acp-8-5327-2008, 2008.

Kesti, J., Asmi, E., O'Connor, E. J., Backman, J., Budhavant, K., Andersson, A., Dasari, S., Praveen, P. S., Zahid, H., and Gustafsson, Ö.: Changes in aerosol size distributions over the Indian Ocean during different

meteorological conditions, Tellus B: Chemical and Physical Meteorology, 72, 1-14, 10.1080/16000889.2020.1792756, 2020.

Kroll, J. H., & Seinfeld, J. H. (2008). Chemistry of secondary organic aerosol: Formation and evolution of low-volatility organics in the atmosphere. Atmospheric Environment, 42(16), 3593–3624. https://doi.org/10.1016/j.atmosenv.2008.01.003

Kroll, J., Donahue, N., Jimenez, J. et al. Carbon oxidation state as a metric for describing the chemistry of atmospheric organic aerosol. Nature Chem **3,** 133–139 (2011). https://doi.org/10.1038/nchem.948

Lamkaddam, H., Dommen, J., Ranjithkumar, A., Gordon, H., Wehrle, G., Krechmer, J., Majluf, F., Salionov, D., Schmale, J., BjeliÄ‡, S., Carslaw, K. S., El Haddad, I., & Baltensperger, U. (2021). Large contribution to secondary organic aerosol from isoprene cloud chemistry. Science Advances, 7(13), 1–11. https://doi.org/10.1126/sciadv.abe2952

Mauritsen, T., Bader, J., Becker, T., Behrens, J., Bittner, M., Brokopf, R., et al. (2019). Developments in the MPI-M Earth System Model version 1.2 (MPI-ESM1.2) and its response to increasing $CO_2$. *Journal of Advances in Modeling Earth Systems*, 11, 998– 1038. https://doi.org/10.1029/2018MS001400

Romakkaniemi, S., Kokkola, H., Lehtinen, K. E. J., and Laaksonen, A.: The influence of nitric acid on the cloud processing of aerosol particles, Atmos. Chem. Phys., 6, 1627–1634, https://doi.org/10.5194/acp-6-1627-2006, 2006.

Stengel, M., Schlundt, C., Stapelberg, S., Sus, O., Eliasson, S., Willén, U., and Meirink, J. F.: Comparing ERA-Interim clouds with satellite observations using a simplified satellite simulator, Atmos. Chem. Phys., 18, 17601–17614, https://doi.org/10.5194/acp-18-17601-2018, 2018.

Tiedtke, M.: Representation of Clouds in Large-Scale Models, Mon Weather Rev, 121, 3040-3061, 10.1175/1520-0493(1993)121<3040:Rocils>2.0.Co;2, 1993.

Tunved, P., Ström, J., and Krejci, R.: Arctic aerosol life cycle: linking aerosol size distributions observed between 2000 and 2010 with air mass transport and precipitation at Zeppelin station, Ny-Ålesund, Svalbard, Atmos. Chem. Phys., 13, 3643–3660, https://doi.org/10.5194/acp-13-3643-2013, 2013.

Tunved, P., Ström, J., and Hansson, H.-C.: An investigation of processes controlling the evolution of the boundary layer aerosol size distribution properties at the Swedish background station Aspvreten, Atmos. Chem. Phys., 4, 2581–2592, https://doi.org/10.5194/acp-4-2581-2004, 2004.